



# Synthesis Product for Ocean Time-Series (SPOTS) – A ship-based biogeochemical pilot

Nico Lange[1], Björn Fiedler[1], Marta Álvarez[2], Alice Benoit-Cattin[3], Heather Benway[4], Pier L. Buttigieg[1,5], Laurent Coppola[6,7], Kim Currie[8], Susana Flecha[9], Makio Honda[10], I. Emma Huertas[11], Siv K. Lauvset[12], Frank Muller-Karger[13], Arne Körtzinger[1,14], Kevin M. O'Brien[15,16], Sólveig R. Ólafsdóttir[3], Fernando C. Pacheco[17], Digna Rueda-Roa[13], Ingunn Skjelvan[12], Masahide Wakita[10], Angelicque White[17,18], Toste Tanhua[1*]

[1]GEOMAR Helmholtz Centre for Ocean Research Kiel, Kiel, Germany
[2]Instituto Español de Oceanografía (IEO-CSIC), A Coruña, Spain
[3]Marine and Freshwater Research Institute, Hafnarfjordur, Iceland
[4]Woods Hole Oceanographic Institution, Woods Hole, Massachusetts, United States
[5]Alfred Wegener Institute, Helmholtz Centre for Polar and Marine Research, Bremerhaven, Germany
[6]Laboratoire d'Océanographie de Villefranche, CNRS, Sorbonne Université, Villefranche-sur-Mer, France
[7]CNRS, OSU STAMAR, UAR2017, Sorbonne Université, Paris, France
[8]National Institute for Water and Atmospheric Research Ltd (NIWA), Dunedin, New Zealand
[9]Instituto Mediterráneo de Estudios Avanzados (CSIC-UIB), Esporles, Spain
[10]Mutsu Institute for Oceanography, Research Institute for Global Change, Japan Agency for Marine-Earth Science and Technology, Mutsu, Japan
[11]Instituto de Ciencias Marinas de Andalucía (CSIC), Cádiz, Spain
[12]NORCE Norwegian Research Centre, Bjerknes Centre for Climate Research, Bergen, Norway
[13]College of Marine Science, University of South Florida, St. Petersburg, Florida, USA
[14]Christian Albrecht University of Kiel, Kiel, Germany
[15]Pacific Marine Environmental Laboratory, National Oceanic and Atmospheric Administration, Seattle, Washington, USA
[16]Cooperative Institute for Climate, Ocean and Ecosystem Studies, University of Washington, Seattle, Washington, USA
[17]Department of Oceanography, University of Hawaii, Honolulu, Hawaii, USA
[18]Daniel K. Inouye Center for Microbial Oceanography: Research and Education, Honolulu, HI, USA

*Correspondence:
Toste Tanhua
ttanhua@geomar.de



**Abstract.** The presented pilot for the "Synthesis Product for Ocean Time-Series" (SPOTS) includes data from 12 fixed ship-based time-series programs. The related stations represent unique marine environments within the Atlantic Ocean, Pacific Ocean, Mediterranean Sea, Nordic Seas, and Caribbean Sea. The focus of the pilot has been placed on biogeochemical essential ocean variables: dissolved oxygen, dissolved inorganic nutrients, inorganic carbon (pH, total alkalinity, dissolved inorganic carbon, and partial pressure of $CO_2$), particulate matter, and dissolved organic carbon. The time-series used include a variety of temporal resolutions (monthly, seasonal, or irregular), time ranges (10 – 36 years), and bottom depths (80 – 6000 m), with the oldest samples dating back to 1983 and the most recent one corresponding to 2021. Besides having been harmonized into the same format (semantics, ancillary data, units), the data were subjected to a qualitative assessment in which the applied methods were evaluated and categorized. The most recently applied methods of the time-series programs usually follow the recommendations outlined by the Bermuda Time-Series Workshop report (Lorenzoni and Benway, 2013) which is used as the main reference for "biogeochemical best-practices". However, measurements of dissolved oxygen and pH in particular, still show room for improvement. Additional data-quality descriptors include precision and accuracy estimates, indicators for data variability, and offsets compared to a reference and widely recognized data product for the global ocean: the "GLobal Ocean Data Analysis Project". Generally, these descriptors indicate a high level of continuity in measurement quality within time-series programs and a good consistency with the GLobal Interior Ocean Carbon Data, even though robust comparisons to the latter are limited. The data are available as (i) a merged comma-separated file that is compliant with the World Ocean Circulation Experiment (WOCE) exchange format and ii) a format dependent on user queries via the ERDDAP server of the Global Ocean Observing System (GOOS). The pilot increases the data utility, findability, accessibility, interoperability, and reusability following the FAIR philosophy, enhancing the readiness of biogeochemical time-series. It facilitates a variety of applications that benefit from the collective value of biogeochemical time-series observations and forms the basis for a sustained time-series living data product, SPOTS, complementing relevant products for the global interior ocean carbon data (GLobal Ocean Data Analysis Project), global surface ocean carbon data (Surface Ocean $CO_2$ Atlas; SOCAT), and global interior and surface methane and nitrous oxide data (MarinE MethanE and NiTrous Oxide product).

Aside from the actual data compilation, the pilot project produced suggestions for reporting metadata, implementing quality control measures, and making estimations about uncertainty. These recommendations aim at encouraging the community to adopt more consistent and uniform practices for analysis and reporting and at updating these practices regularly. The detailed recommendations, links to the original time-series programs, the original data, their documentation, and related efforts are available on the SPOTS website. This site also provides access to the data product (DOI: 10.26008/1912/bco-dmo.896862.1, Lange et al., 2023) and ancillary data.

## 1. Introduction

Continuing global anthropogenic carbon dioxide emissions in combination with increasing nutrient inputs into the ocean over the past decades have resulted in unprecedented changes in the ocean biogeochemistry (O'Brien et al., 2017; Friedligstein et al., 2022) and marine ecosystem states (e.g., Edwards et al., 2013; Barton et al., 2016). As climate change progresses, these complex changes will aggravate (Bopp et al., 2013; Cooley et al., 2022).

To disentangle natural variability, occurring on a range of temporal and spatial scales (Valdés and Lomas, 2017), and human-induced changes in marine ecosystems (Henson et al., 2016; Benway et al., 2019) decades of sustained fixed-location time-series observations are required. Following recommendations from international programs such as the Joint Global Ocean Flux Study (JGOFS, 1990) and Global Ocean Ecosystem Dynamics (GLOBEC, 1997), only few ship-based fixed ocean time-series programs have been established around the globe since the late 1980s. The ongoing observations of these programs have captured the evolving changes in ocean biogeochemistry and associated impacts on marine food webs, marine biodiversity, and ecosystems. Examples of observed changes include changes in the ocean's anthropogenic carbon inventory, oxygen levels, seawater pH, ventilation rates, and vertical nutrient transports (e.g., Bates et al., 2014; Tanhua et al., 2015; Neuer et al., 2017). Even though the collective value of multiple time-series data is greater than that provided by each individual time-series, ship-based time-series programs have primarily been launched to support the specific goals of individual programs and ancillary projects. The International Group of Marine Ecological Time Series (IGMETS, O'Brien et al., 2017) demonstrated the collective value by performing an integrative and collective assessment of over 340 ship-based time-series thereby increasing the range of space- and time scales that can be addressed and highlighting the importance of joint and multidisciplinary time-series observing programs (Valdés and Lomas, 2017).

Despite their indisputable importance and the wealth of ship-based time-series data, difficulties in data discoverability, accessibility, and interoperability presently limit ship-based time-series data utilization, the realization of their full scientific potential, and the overall recognition of the programs (Benway et al., 2019; Tanhua et al., 2021). Moreover, these challenges have prevented shipboard time-series from becoming a more formalized and endorsed component of the Global Ocean Observing System (GOOS, Moltmann et al., 2019). In addition to the lack of a community-agreed time-series data public release agreement that leads to free sharing of time-series data being uncommon, the lack of standardized formats, semantics, units, scales, standards, quality assurance- and control, metadata reporting, and user interfaces across and within time-series sites represent the main data challenges. The usage of different measurement protocols sometimes without comprehensive reporting of the corresponding variable-inherent uncertainties and the time-consuming manual data retrieval at multiple access points are further prone to data handling errors. Existing biogeochemical (BGC) data synthesis products have already tackled these challenges for other observation types and increased the utility of large amounts of individual datasets, e.g., the MarinE MethanE and NiTrous Oxide product (MEMENTO, Kock and Bange, 2015), the Global Ocean Data Analysis Project (GLODAP, Lauvset et al., 2022) and the Surface Ocean $CO_2$ Atlas (SOCAT, Bakker et al., 2016). However, neither IGMETS (O'Brien et al., 2017) nor OceanSites (Weller et al., 2016), a global network of long-term autonomous open ocean reference stations, have generated a global data synthesis product of time-series data that would complement existing BGC data synthesis products.

To address these shortcomings and to follow up on the Bermuda Time Series workshop from 2013 (Lorenzoni et al., 2013), both the Ocean Carbon and Biogeochemistry program and the EU Horizon 2020 project EuroSea convened workshops with several time-series operators. Resulting from these workshops a call was formulated for a pilot data synthesis product of well-established time-series programs that focuses on a limited set of variables. Further, a roadmap was created to develop a pilot product that aims at establishing a Findable Accessible Interoperable Reusable (FAIR, Wilkinson et al., 2016) data management plan for shipboard ocean time-series (Benway et al., 2020). This goes hand in hand with the GOOS Implementation Roadmap (GOOS, 2020) calling for more systematic and sustainable approaches for climate-relevant observations across ocean data platforms and networks (Belward et al., 2016), especially regarding the GOOS defined scientific applications: The ocean carbon content (Q1.1); ocean dead zones (Q2.1); rates of acidification (Q2.2); and ocean productivity (Q3.2).

Following these calls, we here describe the resultant Synthesis Product for Ocean Time-Series (SPOTS) pilot, synthesizing high-quality data from 12 global ship-based time-series sites with a focus on BGC essential ocean variables (EOV). This paper briefly presents the included time-series programs (Sect. 2), describes the methods applied to compile and assess the product (Sect. 3) and data quality assessment (Sect. 4), describes the final product (Sect. 5), elaborates on the stakeholder usability (Sect. 6), and describes the data access (Sect.7). Finally, the main findings of the effort are presented (Sect. 8) and next steps to guarantee the continuity and success of SPOTS are identified (Sect. 9).



## 2. Data Sources

The SPOTS pilot includes data from 12 fixed ship-based time-series programs (Fig. 1), all of which routinely
measure BGC EOVs. All major climate zones are covered, although not all ocean biogeochemical zones are
(Reygondeau et al., 2013). Existing datasets were extended whenever possible by publicly available and more
recent data (Table S1.). In addition to capturing different marine environments (Sect. 2.1), the characteristics of
the time-series programs also differ in terms of the station visit frequency, i.e. temporal resolution (monthly,
seasonal, or irregular), the time range of the observational period, the bottom depth and whether a dedicated
research vessel is used (Table 1). If a time-series program consists of two or more related stations, usually the
deepest station was selected. The included data from GIFT and RADCOR display exceptions to this rule as for
both sites data from three related stations were selected.

**Table 1**: Key metadata of participating time-series programs. Colors indicate ocean basins: Green: Pacific; Light blue: Atlantic;
Orange: Marginal Seas; Dark Blue: Nordic Seas. S=Salinity (either bottle or CTD-data); $O_2$=Oxygen (either bottle or CTD-
data); $NO_3$=Dissolved inorganic nitrate; $NO_2$=Dissolved nitrite; $PO_4$=Dissolved phosphate; $SiOH_4$=Dissolved silicate;
$NH_4$=Dissolved ammonium; DIC=Dissolved inorganic carbon; TA=Total alkalinity; $p$CO$_2$=Partial pressure of carbon dioxide;
POC=Particulate organic carbon; PON=Particulate organic nitrogen; POP=Particulate organic phosphorus; DOC=Dissolved
organic carbon.

| Time-Series Site | Location | Time Range | Temporal Resolution | Bottom Depth | # of Visits | Dedicated Vessel | Variables | Original DOI(s) |
|---|---|---|---|---|---|---|---|---|
| KNOT | 44.0°N 155.0°E | 1997–2020 | 1-3 cruises yr$^{-1}$ | 6000 m | 21 | No | S,$O_2$,$NO_3$,$NO_2$,$SiOH_4$, $PO_4$,$NH_4$,DIC,TA,pH | 10.25921/tarq-6v91 |
| K2 | 47.0°N 160.0°E | 1999–2020 | 1-3 cruises yr$^{-1}$ | 6000 m | 49 | No | S,$O_2$,$NO_3$,$NO_2$,$SiOH_4$, $PO_4$,$NH_4$,DIC,TA,pH, DOC | 10.25921/mpfz-sv16 |
| ALOHA | 22.8°N 158.0°W | 1988–2019 | Monthly | 4750 m | 311 | Yes | S,$O_2$,$NO_3$,$NO_2$,$SiOH_4$, $PO_4$,DIC,TA,pH, POC,PON,POP,DOC | 10.1575/1912/bco-dmo.3773.1 |
| Munida | 45.8°S 171.5°E | 1998–2019 | 6 cruises yr$^{-1}$ | 1000 m | 80 | Yes | S,$NO_3$,$SiOH_4$,$PO_4$,DIC ,TA | NA |
| GIFT | 36.9°N 6.0°W | 2005–2015 | Seasonal | 315 m – 842 m | 26 | Yes | S,$O_2$,$NO_3$,$SiOH_4$,$PO_4$, TA,pH, DOC | 10.20350/digitalCSI C/10549 |
| CVOO | 17.6°N 24.3°W | 2006–2019 | 1-3 cruises yr$^{-1}$ | 3600 m | 42 | Partly | S,$O_2$,$NO_3$,$NO_2$,$SiOH_4$, $PO_4$,$NH_4$,DIC,TA, POC,PON,POP | 10.1594/PANGAEA .958597 |
| RADCOR | 43.4°N 8.4°E | 2013–2020 | Monthly | 15 m – 80 m | 80 | Yes | S,$O_2$,$NO_3$,$NO_2$,$SiOH_4$, $PO_4$,DIC,TA,pH | NA |
| CARIACO | 10.5°N 64.7°W | 1995–2017 | Monthly | 1300 m | 230 | Yes | S,$O_2$,$NO_3$,$NO_2$,$SiOH_4$, $PO_4$,$NH_4$,TA,pH, POC,PON,POP,DOC | 10.1575/1912/bco-dmo.3093.1 |
| DYFAMED | 42.3°N 7.5°E | 1991–2017 | Monthly | 2400 m | 190 | No | S,$O_2$,$NO_3$,$NO_2$,$SiOH_4$, $PO_4$,,DIC,TA,pH | 10.17882/43749 |
| IrmingerSea | 64.3°N 28.0°W | 1983–2019 | Seasonal | 1000 m | 131 | Yes | S,$O_2$,$NO_3$,$SiOH_4$,$PO_4$, DIC,TA,$p$CO$_2$ | 10.3334/cdiac/otg.ca rina_irmingersea_v2 ; 10.25921/vjmy-8h90 |
| IcelandSea | 68.0°N 12.7°W | 1983–2019 | Seasonal | 1850 m | 146 | Yes | S,$O_2$,$NO_3$,$SiOH_4$,$PO_4$, DIC,TA,$p$CO$_2$ | 10.3334/cdiac/otg.ca rina_icelandsea; 10.25921/qhed-3h84 |
| OWSM | 66.0°N 2.0°E | 2001–2021 | 4-12 cruises yr$^{-1}$ | 2100 m | 147 | Until 2009 | S,$O_2$,$NO_3$,$SiOH_4$,$PO_4$, DIC,TA | 10.3334/cdiac/otg_ts m_ows_m_66n_2e |



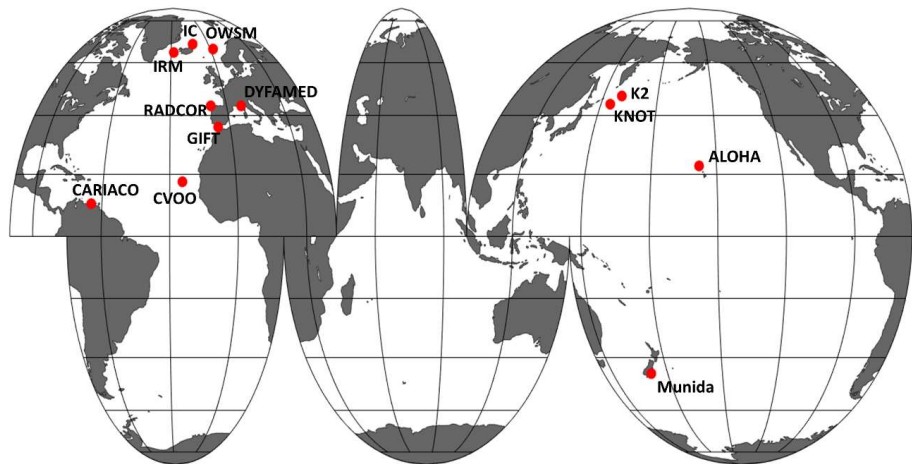

**Figure 1**: Locations of participating ship-based time-series stations.

### 2.1. Marine Environment of Time-Series Sites

#### 2.1.1. A Long-term Oligotrophic Habitat Assessment (ALOHA)

The deep water (~4750 m) time-series station of the Hawaii Ocean Time-Series program (HOT), ALOHA (Karl and Church, 2019), is located 100 km north of Oahu, Hawaii, more than one Rossby radius (50 km) away from the steep topography associated with the Hawaiian Ridge. ALOHA serves as an open ocean benchmark and its research goals are aligned with the main objectives of the JGOFS and the World Ocean Circulation Experiment (WOCE). One of the principals of the HOT program is to observe seasonal and interannual variations in water

mass characteristics and BGC variables. The monthly measurements since 1988 are representative of the oligotrophic North Pacific eastern subtropical gyre with Station ALOHA lying in the center of the North Pacific and North Equatorial Current. Typically, the site is characterized by a relatively deep permanent pycnocline (and nutricline), and a shallow mixed-layer depth. Intermittent local wind forcing caused by extratropical cyclones' cold fronts impacts the annual cycle of the surface waters (Karl et al., 1996).

#### 2.1.2. CArbon Retention In A Colored Ocean (CARIACO)

The station of the CARIACO Oceanographic Time-Series Program (Muller-Karger et al., 2019) is located in the Cariaco Basin, a semi-enclosed tectonic depression located on the continental shelf off northern Venezuela in the southern Caribbean Sea. The Cariaco Basin is composed of two approximately 1400 m deep sub-basins that are

160 connected to the Caribbean Sea by two shallow (140 m deep) channels. These channels allow for the open exchange of near-surface water. The restricted circulation below the 140 m sills, coupled with highly productive surface waters due to seasonal wind-driven coastal upwelling (around 450 g C m$^{-2}$ y$^{-1}$; Muller-Karger et al., 2010), has led to sustained anoxia below around 250 m. The goal of the near-monthly measurements at CARIACO between 1995 and 2017 was to observe linkages between oceanographic processes and the production,

remineralization, and sinking flux of particulate matter in the Cariaco Basin, and how these change over time. It also aimed at understanding climatic changes in the region.

#### 2.1.3. Cape Verde Ocean Observatory (CVOO)

CVOO is located in the eastern tropical North Atlantic about 800 km from the west coast of Africa, which is

170 influenced by the seasonal eastern boundary upwelling system, high Saharan dust deposition rates, and frequently passing eddies (Schütte et al., 2016). It is part of the Cape Verde Observatory, which also includes an operational atmospheric monitoring site. The combined observations aim at investigating long-term changes of greenhouse gas concentrations in the atmosphere and in the ocean in a key region for air-sea interaction. The irregular measurements of BGC variables at CVOO started in 2006 and are still ongoing, and the project strives for more

regular measurements in the future by having a dedicated vessel available. The station has a bottom depth of 3600 m and lies in the center of the Cape Verde Fontal Zone, resulting in large variations of the present oligotrophic water masses. The frontal zone separates most of the eastern tropical North Atlantic from the anticyclonic subtropical gyre system in the North Atlantic (Stramma et al., 2005). This further results in an ocean shadow zone



and an oxygen-poor layer between 400 m to 500 m (Stramma et al., 2008), which is being sampled at CVOO.
Below the mixed layer, subtropical underwater from the subtropical gyre system, as well as North Atlantic Central
Water and South Atlantic Central Water can be present (Tomczak 1981; Pastor et al., 2008).

### 2.1.4. DYFAMED

DYFAMED is located in the central part of the Ligurian Sea, about 50 km off Nice, on the Nice Corsica transect,
and is representative of open sea western Mediterranean basin waters. Ongoing multidisciplinary monthly
measurements at DYFAMED have been performed since 1991 observing: i) the evolution of the water mass
properties, ii) the carbon export change, and iii) the variability of the biological species relative to climate forcing.
The water column can be divided into three principal layers: deep, intermediate, and surface. The latter, typically
for the Mediterranean trophic environment, experiences large seasonal variability. Further, the Northern Current
front acts as a barrier to exchanges with the coastal zone of the Ligurian Sea and prevents DYFAMED from lateral
inputs (Vescovali et al., 1998). Consequently, the primary production depends on inputs of nutrients from deeper
waters and atmospheric inputs of nitrogen and some trace metals, particularly during summer (Miquel, 2011). The
DYFAMED site is characterized by intermediate water (300-400m) that is lower in oxygen concentrations
(Levantine Intermediate Water) and deep water that is richer in oxygen, primarily induced by vertical mixing
occurring in winter during intense and cold winds (convection processes; Coppola et al., 2018).

### 2.1.5. Gibraltar Fixed Time series (GIFT)

Seasonal measurements at GIFT were established in 2005 to quantify the exchange of carbon between the
Mediterranean Sea and the adjacent Atlantic Ocean and assess the temporal evolution of BGC fluxes. The three
GIFT time-series stations (Flecha et al., 2019) are located along the longitudinal axis of the Strait of Gibraltar,
which connects the two basins. The Strait is surrounded by the Gulf of Cadiz (west) and the Alboran Sea (east).
Water circulation in the channel can be described as a bi-layer system characterized by an inward (eastward) flow
of the North Atlantic Central Water in the upper layer and an outward (westward) flow of Mediterranean waters
(predominantly formed by a mixture of the Levantine Intermediate Water and the Western Mediterranean Deep
Water) at the bottom layer. The depth and thickness of each water mass vary along the Strait, due to topography
in the channel and the influence of physical mechanisms. In particular, the Espartel sill (358 m depth) and the
Camarinal sill (285 m depth) lead to large variability in the proportion of water flows' position. Therefore,
sampling depths vary from one campaign to another due to the instant position of the incoming and outcoming
flows that are identified by their thermohaline properties through the CTD casts.

### 2.1.6. Irminger Sea station (IRM-TS) and Iceland Sea station (IC-TS)

In 1983, seasonal measurements at the IRM-TS and the IC-TS (Olafsson et al., 2010) were initiated to observe the
seasonal variability of carbon-nutrient chemistry in the North Atlantic off the Iceland shelf. The stations are located
in two hydrographically different regions north and southwest of Iceland (Takahashi et al., 1985; Peng et al., 1987).
The station in the northern Irminger Sea (IRM-TS) is characterized by relatively warm and saline (S > 35) Modified
North Atlantic Water derived from the North Atlantic Drift. Winter mixing is induced by strong winds and loss of
heat to the atmosphere. This location may also be described as representing the subpolar gyre (Hatún et al., 2005).
The IS-TS is located in the central Iceland Sea north of the Greenland-Scotland Ridge. At the IC-TS cold Arctic
Intermediate Water, formed from Atlantic Water and low salinity Polar Water, usually predominates and overlays
Arctic Deep Water (Olafsson et al., 2009). The Polar Water influence in the surface layers is variable (Stefansson,
1962; Hansen and Østerhus, 2000). Both regions are important sources of North Atlantic Deep Water.

### 2.1.7. K2 and KNOT

The K2 and KNOT stations (Wakita et al., 2017) are located approximately 400 km northeast of Hokkaido Island,
Japan in the subarctic western North Pacific. Since 2001 and 1997, respectively, irregular field observations have
been conducted at these stations to investigate the inorganic carbon system dynamics in response to variations in
hydrography and biological processes. The overarching goal is to investigate the response of the biological pump
to climate forcing in the western subarctic Pacific gyre. The region is characterized by high primary productivity,
abundant marine resources (FAO, 2016) and might be the first region of the ocean to become undersaturated with
respect to calcium carbonate during winter (Orr et al., 2005). The sites are representative of the southwestern
subarctic gyre with both stations lying offshore of the Oyashio Current and just north of the subpolar front.
Seasonal cycles are present (e.g., Takahashi et al., 2006; Tsurushima et al., 2002; Wakita et al., 2013) with a highly
productive biological pump from spring to fall and strong vertical mixing of deep waters that are rich in dissolved
inorganic carbon (DIC) in winter.




### 2.1.8. Munida

This deep-water station is located in the Southwest Pacific Ocean 65 km off the southeast coast of New Zealand and is part the Munida Time Series Transect, which is sampled every two months. Measurements at Munida were established in 1998 to study the role of these waters in the uptake of atmospheric carbon dioxide, and the seasonal,
interannual, and long-term changes of the carbonate chemistry. The subantarctic waters are a sink for atmospheric carbon dioxide (Currie et al., 2011), and the seasonal cycles of DIC are primarily driven by net community production (Brix et al, 2013; Jones et al., 2013) with modification by the annual cycle of sea surface temperature.

### 2.1.9. Ocean Weather Station Mike (OWSM)

OWSM is located in the Norwegian Sea at the western baroclinic branch of the northwards-flowing Norwegian Atlantic Current where the water depth is 2100 m (Skjelvan et al., 2008; 2022). Hydrographic measurements date back to 1948 while carbonate chemistry measurements started in 2001 to monitor long-term changes in the biogeochemistry. Between 2001 and 2009, the station was sampled monthly, and since 2010, the sampling frequency has been four to six times per year. The site encompasses the cold Norwegian Sea Deep Water and the
Arctic Intermediate Water in addition to the relatively warm and saline Atlantic Water. Occasionally during late summer, fresh Norwegian Coastal Current Water meanders all the way out to OWSM, influencing the surface water at the station. Seasonal variability is observed in the uppermost ~200 m, and long-term trends of carbonate variables are observed at all water depths. Over time, the surface water $CO_2$ content at OWSM has increased at a faster rate than atmospheric $pCO_2$ at this site (Skjelvan et al., 2022).


### 2.1.10. A Coruña RADIALES (RADCOR)

The RADIALES program started in 1989 aiming to obtain reliable baselines for long-term studies on climate change and ecosystem dynamics in times of increasing anthropogenic disturbances along the northern and northwestern Spanish coasts (Valdés et al., 2021). The program consists of monthly multidisciplinary
perpendicular sections covering the Cantabrian Sea and northwest coastal and neritic Spanish ocean. The A Coruña (NW Galician coast) section (RADCOR) started in 1990 (Bode et al., 2020) and $CO_2$ variables have been incorporated since 2013 in two stations, E2CO and E4CO. RADCOR is located on the northern edge of the Iberian Upwelling Region. Here, the classical pattern of seasonal stratification of the water column in temperate regions is masked by upwelling events from May to September. These upwelling events provide nutrients to support both
primary and secondary production in summer. Nevertheless, upwelling is highly variable in intensity and frequency, demonstrating substantial interannual variability, mostly affecting the E2CO station (80 m), while the station closest to shore, E4CO (15 m), is more impacted by estuarine and benthic processes.

## 3. Methods

The data flow of the SPOTS pilot depicting the main steps of the synthesis is schematically illustrated in Fig. 2. In the following, the individual components of this data flow are described in detail.

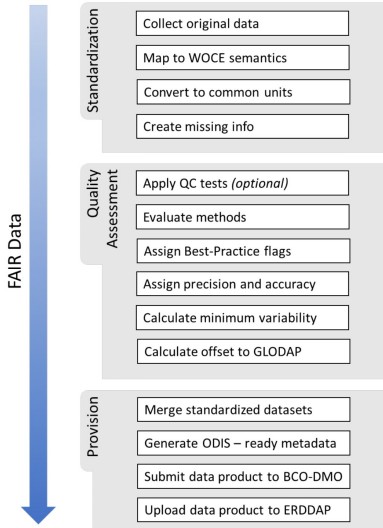

**Figure 2**: Schematic data flow of the SPOTS pilot

### 3.1. Data Collection

The data from the 12 participating time-series programs were retrieved from data centers or directly obtained from the responsible principal investigator (Table S1). In the latter case, merging, formatting, additional quality-control (QC), and archiving of existing data were carried out. Only bottle data for BGC EOVs, that had been measured by at least two of the participating programs were included in the pilot project, along with accompanying ancillary pressure, salinity, and temperature data. We have also developed a metadata template for BGC EOV ship-based

time-series data (Table S2). The template has subsequently been used to collect all relevant metadata information from each participating time-series program. The collected metadata includes general information about the program, such as information about the principal investigator and the location and timeframe of related station(s). It also includes detailed information on the measured variables - e.g., units; sampling and analytical methods and associated instrumentation; calculation, calibration, and quality control procedures; and standards or (certified)

reference materials used. The latter not only vary among the time-series programs, but can also vary within a time-series program over time.

### 3.2. Data Assembly

The SPOTS pilot was created by standardizing data format, units, header names, primary QC flags, times,

locations, and fill values and subsequently merging the individual datasets of each time-series program into one file. Only data that received a WOCE quality flag 2 (Table S3) were included in the product. Existing data were altered as little as possible without interpolation or calculation of "missing" variables. Similarly, original station-, cast- and bottle numbers were kept or created artificially if non-existent to ensure consistency. The headers, units, and flags of the individual time-series datasets were standardized (Table S4) to conform with the WOCE exchange

bottle data format (Swift and Diggs, 2008), a comma-delimited ASCII format for bottle data from hydrographic cruises. To enable an automated mapping to other existing vocabularies, we also mapped the WOCE headers to the Natural Environment Research Council (NERC) British Oceanographic Data Centre P01 vocabulary collection, as well as to the newly proposed BGC bottle standard by Liqing et al. (2022). We did not use the latter as "central" semantics due to restrictions of existing QC-tools (e.g., AtlantOS QC (Velo et al., 2022) and the

crossover toolbox (Tanhua et al., 2010; Lauvset and Tanhua, 2015)) to WOCE semantics.

The standardization process also entailed unit conversions, most frequently from micromoles per liter ($\mu$mol L$^{-1}$; nutrients and dissolved organic carbon (DOC)) or from micrograms per kilogram ($\mu$g kg$^{-1}$; particulate matter) to micromoles per kilogram of seawater ($\mu$mol kg$^{-1}$). The default procedure to convert from volumetric to gravimetric

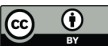

units was to use seawater density at in-situ salinity, reported laboratory temperature (otherwise assuming 20°C as laboratory conditions), and pressure of 1 atm (following recommendations from Liqing et al., 2022). For some time-series datasets, the combined concentration of nitrate and nitrite was reported (Table S4). If explicit nitrite concentrations were provided, these were subtracted to obtain the nitrate values. If not, the combined concentration was renamed to nitrate assuming that the relative nitrite amount is negligible. For the HOT program specifically, low-level, high-sensitivity measurements of macronutrients (phosphate and nitrate) were available but not included in the pilot product. Particulate organic matter was derived by subtracting the particulate inorganic matter from the total particulate matter, if available. For particulate organic carbon and particulate organic nitrogen, the factors 1/12.01 and 1/14.01 (inverse standard atomic masses) were used, respectively, for the unit conversion to micromoles per kilogram. For the HOT program, particulate carbon and nitrogen measurements correspond to total particle concentrations (PC and PN), but are here assumed to approximate POC and PON. If neither temperature nor pressure was provided, all corresponding data entries were excluded from the product. The potential density anomaly[1] is the only calculated variable. Missing and excluded values were set to -999.

### 3.3. Qualitative Assessment of Data
### 3.3.1. Internally Applied Quality-Control (QC)

The majority of the programs have established their own routines for QC and correspondingly flag their data using different flagging schemes. We did not double-check the applied flags, nor did we run additional QC checks. The applied QC on the collected stations include statistical outlier checks on routinely measured pressure intervals using either a two- or three- (seasonal) sigma criteria, visual inspections of property-property plots (PPP), and application of crossovers using reference layers (Table S5). For example, K2 and KNOT used North Pacific Deep Water (NPDW), defined as the water mass between 27.69 $\sigma_\theta$ (around 2000 dbar) and 27.77 $\sigma_\theta$ (around 3500 dbar) (Wakita et al., 2017), as the reference layer for their internal crossover checks. For CVOO and Munida, we performed QC by applying a seasonal two-sigma criterium to the data, and for CVOO, we made additional use of comparisons to CANYON-B (Bittig et al., 2018) and crossovers. Since the QC procedures differ from program to program, we have provided recommendations for the QC of future data, so that the flags are applied more consistently across different programs (Sect. 6.3). Further, the standardization of the SPOTS pilot also entailed mapping to a central flagging scheme. We chose the WOCE bottle flag scheme (Table S3). Flags indicating replicate measurements (WOCE flag of 6) were set to 2, whereas all other flags were set to 9 and the corresponding values to -999.

### 3.3.2. Best-Practices (BP) Assessment

Given the inconsistencies in the applied internal quality checks and the fact that bias corrections following crossovers analyses are presently impossible to apply to all included time-series datasets[2], the comparability of the data for the SPOTS pilot was qualitatively assessed. The information on the applied methods of each time-series program, as provided through the metadata collection, was evaluated against, ideally, published Best Practices (BPs), and otherwise known standard operating procedures (SOPs). "BP Flags" were assigned accordingly to each cruise of a time-series program (Table 2).

**Table 2**: Meaning of assigned BP Flags.

| Flag | Definition |
|------|------------|
| **0** | No data |
| **1** | Methods meet all BP requirements (including "desired") |
| **2** | Methods only meet "required" BP requirements |
| **3** | Methods do not meet the BP requirements (or no metadata given) |

The majority of the defined "BP requirements" used for the evaluation are based on the Bermuda Time-Series Workshop report (Lorenzoni and Benway, 2013), with additional implementation of: GO-SHIP manuals (Langdon et al., 2010; Becker et al., 2019); the CARIACO Methods Manual (Astor et al., 2013); HOT analytical methods (https://hahana.soest.hawaii.edu/hot/protocols/protocols.html), which are based on the Joint Global Ocean Flux

---

[1] Calculated using the Matlab seawater toolbox (Morgan, 1994)
[2] Crossover require a "constant" reference layer over the entire span of measurements. Especially in coastal and shallow water formation regions this layer is nonexistent. Detrending might make this criterion redundant. However, detrending techniques rely on regular measurement intervals, which is not the case for most ship-based time-series sites.



Study protocols (IOC, 1994); the guide to BPs for ocean $CO_2$ measurements (Dickson et al., 2007); results from the Scientific Committee on Research Working Group 147 "Towards comparability of global oceanic nutrient data" (Bakker et al., 2016; Aoyama et al., 2015); and studies on preservation techniques for nutrients (e.g. Dore et al., 1996). The requirements were grouped into "Required" and "Desired" BP, see Table 3. To fulfill all requirements, i.e. receive a BP flag of 1, the metadata must show that the methods also met the corresponding "Desired" requirements. Only time-series programs that provided granular metadata, i.e. metadata differentiating between different methods applied in time, could obtain a BP flag of 1.

**Table 3**: BP requirements used for the method evaluation.

| Variable | | Required | Desired |
|---|---|---|---|
| **Salinity** | | AutoSal | (Sub-) standard used regularly |
| | | | Temperature constant |
| | | | Glass bottles |
| **Dissolved Oxygen** | | Winkler | Draw temperature used for mass calculation if difference to in situ temperature > 2.5°C |
| | | | Titration reagent assessed using CSK/OSIL primary standard |
| **Nutrients** | **All** | Autoanalyser; If stored: Frozen upright | Carrier Solution documented |
| | | | Calibrated against Reference Material |
| | **Silicate** | Autoanalyser; If stored and concentrations are above 40 µmol $L^{-1}$: Poisoned and kept cold | Carrier Solution documented |
| | | | Calibrated against Reference Material |
| **Dissolved Inorganic Carbon** | | Coulometry | Calibrated against Certified Reference Material (Andrew Dickson, SIO) |
| | | | If stored: Poisoned, kept in dark and cool location |
| | | Potentiometric (closed-cell); Calibrated against Certified Reference Material (Andrew Dickson, SIO); If stored: Poisoned and kept in a cold, dark location[3] | Not applicable |
| **Total Alkalinity** | | Potentiometric Titration (multi-step) | Open Cell or curve fitting method documented |
| | | | Calibrated against Certified Reference Material (Andrew Dickson, SIO) |
| | | | If stored: Poisoned, kept in dark and cool location |
| | | Spectrophotometric | Indicator dye: bromocresol green |
| | | | Calibrated against Certified Reference Material (Andrew Dickson, SIO) |
| | | | If stored: Poisoned, kept in dark and cool location |
| **pH** | | Spectrophotometric with scale and temperature reported | Indicator dye: m-cresol purple |
| | | | Indicator dye: Purified |
| | | | If dye is not purified: Correction applied to impurities |
| **Partial pressure of $CO_2$** | | Gas-chromatography | Temperature and standard reported |
| | | | If stored: Poisoned, kept in dark and cool location |
| | | Infrared-based system | Temperature and standard reported |
| | | | If stored: Poisoned, kept in dark and cool location |
| **Particulate Matter** | **Carbon and Nitrate** | High temperature combustion with reported filter volume and pore size | Dried filters (60°C) |
| | | | Standards reported |

---

[3] Capped at a BP flag 2.



| | Phosphorus | Ash hydrolysis with reported filter volume and pore size | Dried filters (60°C) |
|---|---|---|---|
| | | | Standards reported |
| **Dissolved Organic Carbon** | | High temperature combustion | Filtered |
| | | | If stored: Frozen or acidified to and refrigerated |
| | | | Calibrated against Reference Material (Dennis Hansell, University of Miami) |

### 3.4. Quantitative Assessment of Data

In addition to the qualitative BP assessment (Sect. 3.3), the bottle data of the time-series are described by our quantitative descriptors: 1) precision, 2) accuracy, 3) variability on the most consistent depth layer, and 4) consistency with GLODAP (Lauvset et al., 2022). Precision and accuracy were included in the SPOTS pilot dataset file, the latter two were not included and are only described here.

### 3.4.1. Precision and Accuracy

Precision and accuracy estimates, as provided by each time-series program's primary quality-assurance procedure, were assigned to the bottle data. The temporal resolution of these estimates varies from estimates given for each cruise, i.e. on a cruise-to-cruise basis, to estimates given for longer time periods (covering multiple cruises) without recorded changes in applied methodology (Table S6), depending on the individual time-series' internal procedure. If only one estimate was given for a variable for the entire time-series, that estimate was only assigned to the most recently applied method. The units correspond to the units of the respective variable.

Precision estimates are based on replicate samples and expressed as one standard deviation of the replicate measurements[4]. For the carbon variables, the assigned accuracy estimates represent the deviation from certified reference materials from the A. Dickson Laboratory (Scripps Institution of Oceanography). The pH accuracies of RADCOR are an exception, representing the difference from the theoretical TRIS buffer value at 25°C. For oxygen concentrations, the assigned accuracy estimates represent the accuracy of the $KIO_3$ primary standard normality assessed using a certified reference standard from either Ocean Scientific International Ltd (OSIL) or Wako Pure Chemical Industries (WAKO). For nutrient concentrations, the assigned accuracy estimates represent the deviation from reference material from either OSIL, WAKO, or QUASIMEME (Wells et al., 1997) or from certified reference material from Kanso Technos Co., Ltd. (KANSO). For particulate phosphorus concentrations, the assigned accuracy estimates represent deviations from National Institute of Science and Technology (NIST) apple leaves (0.159% P by weight). For DOC, the accuracy estimates represent deviations from deep seawater reference material from D. Hansell (RSMAS, University of Miami). The exact calculations to express the above deviations from reference materials differ slightly across the time-series programs (Table S7), thereby preventing combined precision and accuracy estimates to calculate a total uncertainty in a consistent manner. The estimates should not be confused with values provided by instrument manufacturers, which are ideal values and are usually well below real-world uncertainties.

### 3.4.2. Minimum Variability

To provide an internal consistency measure of measurement quality, we determined the minimum variability of each BGC variable for each time-series station on the pressure surface (+/- 100 dbar) with the least oxygen variability, i.e. the layer on which oxygen has the lowest coefficient of variation. We chose oxygen as natural variability in oxygen can be linked to either variation in ventilation, water mass changes, or changes in consumption and production by biological activity[5] (Sarmiento and Gruber, 2006; Keeling et al., 2010; Stramma and Schmidtko, 2019). As these natural oxygen changes are likely to be accompanied by changes in other BGC variables, we used the layer that is closest to an oxygen equilibrium as an approximation for the least natural variability in ocean BGC. In addition, this choice allowed us to use the salinity variability as an independent indicator of natural variability. For i) CARIACO, ii) GIFT, iii) Munida, and iv) RADCOR, this layer could not be determined properly, respectively, due to i) anoxic water masses below the mixed layer, ii) varying measurement depths, iii) no oxygen data and iv) a shallow bottom depth of 80 m. The minimum variabilities of the other variables

---

[4] Exception: IRM- and IC TS using $V_{dub}*C_{mean}$ (following OSPAR, 2011), where $V_{dub}$ is coefficient of variation calculated from dublicates and $C_{mean}$ is the mean of the concentration measured.

[5] Not represented in the variability of salinity



were subsequently determined by calculating the coefficient of variation of all samples on the identified pressure surface. A minimum of 10 samples on the pressure surface was required.

### 3.4.3. Comparisons to GLODAP

The final quantitative descriptor indicates how well the time-series data compares to the GLODAP dataset (GLODAPv2.2021, Lauvset et al., 2021) and vice-versa, with no a prioi assumption of which is 'correct'. To this end, we applied an adapted version of the GLODAP crossover routine to all individual cruises of the time-series programs. Generally, the crossover routine calculates a depth-independent offset between a cruise and a reference dataset based on multiple crossing cruises, i.e. "crossover-pairs". The secondary quality control of GLODAP depends heavily on this routine to determine and correct for biases of new cruises, which results in the high internal consistency of the core GLODAP variables. In the following, we first describe the crossover routine of GLODAP in detail and subsequently highlight the modifications applied to the routine so that it fits our pilot product needs. For a given variable the depth-independent offset of a new cruise against GLODAP is calculated using the following steps:

Step 1) Detect all GLODAP cruises that cross the to-be-compared cruise (denoted as Cruise A in the following), i.e. find all "crossover-pairs" of Cruise A in GLODAP. In the 2nd QC of GLODAP, a "crossover-pair" is defined by two cruises that have (at least) three stations within a 2° radius that include (at least) three samples below a minimum of 1500 dbar. These requirements ensure that the influence of natural signals on the calculated offsets is limited. That becomes especially important if the time period between Cruise A and a crossing GLODAP cruise (denoted as Cruise B in the following) is large.

Step 2) Interpolate the samples of Cruise A and Cruise B to the same standard depths. Usually, the concentrations are compared on sigma-4 surfaces[6]. Samples above the chosen minimum depth are ignored to exclude layers that are influenced by daily to interannual variability.

Step 3) Compare all existing samples of Cruise A and B that are on the same depth surface and from stations within 2°. For each depth surface, the individual offsets are averaged to obtain depth-dependent mean offsets and standard deviations. For nutrients and oxygen, the offsets are multiplicative, and for the carbon variables and salinity, the offsets are additive.

Step 4) Calculate the constant offset of Cruise A against Cruise B by inverse variance weighting all depth-dependent offsets. The resultant depth-independent offset is also known as the crossover-pair offset.

Step 5) Calculate the standard deviation of the crossover-pair offset by inverse variance weighting all depth-dependent standard deviations. This crossover-pair standard deviation reflects the similarity of the offsets within one depth surface and across all depth surfaces. The lower it is, the higher the confidence in the crossover-pair offset.

Step 6) Repeat Steps 2) to 5) for all identified crossover-pairs.

Step 7) Calculate the total offset of Cruise A against GLODAP by inverse variance weighting all calculated crossover-pair offsets. The resultant standard deviation describes the overall uncertainty in the total offset.

For our purposes, we applied an adapted version of the above-described crossover routine using GLODAP as the "reference dataset" against which each time-series station is compared. The term "reference dataset" does not imply that the quality of GLODAP is higher than the quality of the time-series programs, only that it represents a dataset with known consistency in time and space. Each cruise of a time-series station, i.e. station visit, represents another Cruise A in the above-outlined crossover steps.

For a given time-series station and variable, our adapted crossover routine starts with the identification of crossover-pairs for each station visit, similar to Step 1. However, since multiple time-series cruises only take one profile with fewer than three samples below 1500 dbar, we could not apply the same crossover-pair requirements. We kept the distance requirement of 2° and added a new temporal requirement, that only crossover cruises within +/- 45 days were included in the routine. That permitted relaxing the minimum depth requirement and dropping

---

[6] In regions with a high probability of internal waves, in upwelling and water formation regions the offsets are calculated on pressure surfaces.



the requirement of the minimum number of profiles. Note that we excluded crossover-pairs of cruises that are included in both products (parts of: IC-TS, IRM-TS, and OWSM). Steps 2 to 6 of the routine are identical and repeated for all time-series station cruises. In the next step, all crossover pair offsets against the same GLODAP cruise, i.e. a particular Cruise B, are averaged. This step was necessary when multiple time-series cruises took place within 90 days and all were compared to the same Cruise B. Consequently, we obtained one depth-independent offset (and standard deviation) of the time-series station against each GLODAP cruise that meets the crossing requirements. In a final calculation, we determine the total offset of the time-series station against GLODAP by inverse variance weighting all obtained time-series station offsets. If the standard deviation of the time-series station offset against a particular cruise B was below the consistency estimates of GLODAPv2 (see Table 11 in Olsen et al., 2016), the latter ones were used as standard deviations (e.g. only one crossover pair exits between the entire time-series and a particular Cruise B). The routine was only applied to variables defined as core variables[7] in GLODAP. Negative (or lower than unity) offsets indicate lower values compared to GLODAP and vice versa.

---

[7] Salinity, oxygen, nitrate, phosphate, silicate, DIC, total alkalinity and pH

## 4. Data Assessment Results

### 4.1. Best-Practices (BP) Evaluation

The results of the BP assessment indicate that the time-series programs have documented their methodology well and that the most recent methods generally follow BPs (Fig. 3 and Table 3). The proportion of data allocated a BP flag 1 is strongly dependent on the variable and program assessed. The assigned flags partly reflect that over the 40 years multiple method changes occurred (Fig. 4). Method changes are even more pronounced in programs without a dedicated vessel (Table 1). However, not all changes are captured by the assigned BP flags, e.g. instrument changes (Table S6). Note that the overall percentages in Fig. 3 are skewed towards ALOHA, as the number of ALOHA samples makes up around 60% of all samples of the product (Sect. 5).

Further, note that BPs are constantly evolving and consequently this assessment must be seen as "dynamical". In some cases, programs explicitly choose to not follow the most recent recommendations in favor of method consistency. E.g., unpublished internal analyses and discussions in the HOT program about possible advantages and disadvantages of a purified dye for the pH measurements (recommended following the Bermuda Time-Series Workshop report) resulted in not changing their dye. These additional analyses demonstrate the difficulties in determining BPs, but the knowledge is often not shared with the wider community. Hence, regular time-series workshops that discuss currently applied methodologies, achieve community consensus, and result in BP recommendations that are implemented accordingly in the here applied assessment, should take place regularly.

In the following, the results will briefly be presented for each assessed variable.

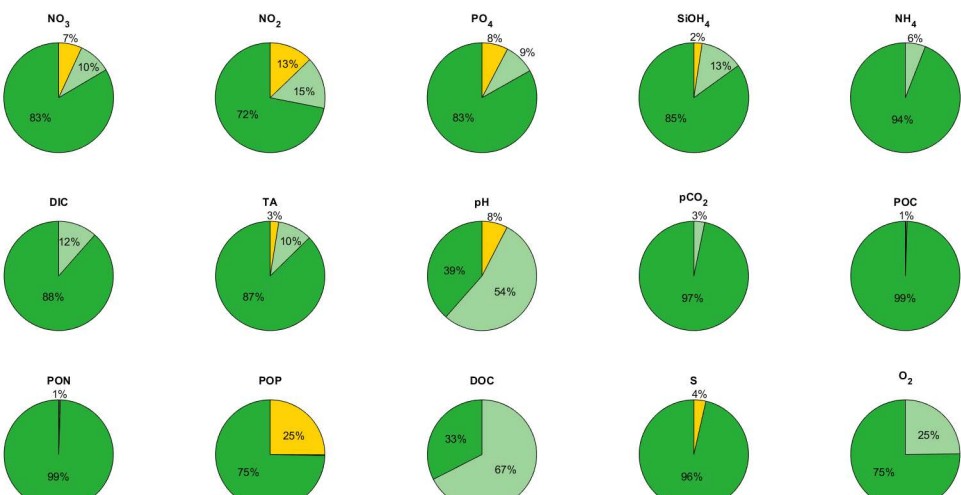

**Figure 3**: Overview of assigned BP flags. Percentages correspond to the number of samples in the combined dataset. Dark green colors indicate samples that have been measured according to all (incl. "desired") BP requirements, i.e. a BP flag 1 (Table 2). Light green colors indicate samples that have been measured meeting the "Required" BP requirements only, i.e. a BP flag 2. Orange colors indicate samples for which the methods do not meet the BP requirements, i.e. a BP flag 3. Variable synonyms correspond to the product header names (Table S8).

### 4.1.1. Salinity

For salinity, 96% of the bottle samples meet all BP criteria. DYFAMED, GIFT, Munida, and RADCOR only provided CTD salinity values and are not included in this statistic. The remaining 4% of bottle salinity samples with a BP flag 3 are a few cruises from ALOHA and CVOO. Salinity samples of the first 26 cruises of ALOHA were measured using an AGE Minisal 2100 salinometer. Also, the first 23 cruises of ALOHA used plastic bottles (instead of glass bottles) to sample salinity, which made them more prone to evaporation. Note that the data were corrected for it. Further, measurements taken on CVOO's research vessel "Islandia" used a Micro-Salinometer MS-310 (RBR Ltd., Canada) instead of a required AutoSal (Guideline Instruments, Canada).

### 4.1.2. Oxygen

Even though the overall statistics show that 75% of all bottle oxygen samples were measured according to the required BPs, 6 out of the 11 programs (Munida time-series program does not measure oxygen) did not regularly

use certified reference $KIO_3$ (CSK, WAKO, OSIL) to assess the accuracy of the Winkler titration measurements. ALOHA, DYFAMED, GIFT, K2, KNOT, and RADCOR (as well as very few cruises from CVOO) used standard reference iodate. Further, note that during the first 10 HOT cruises, the in-situ temperature was used to calculate the mass, rather than the sample draw temperature, resulting in a slightly negative bias which is reflected in a BP Flag 2 of the concerned oxygen samples. The Winkler end-point detection method was either visual (starch) or computer-controlled potentiometric detection, both of which are accommodated in the applied BP assessment.

### 4.1.3. Nutrients

In most cases, all nutrient variables were measured simultaneously using one water sample (and/or with replicates at a single depth sampled), and the applied methods were identical. This is represented in similar BP flags of the nitrate, phosphate, and silicate samples. For these three variables, around 95% of the applied methods met either all BP requirements or the "required" requirements. The most restricting BP requirement is the comparison to reference materials, which, especially for older datasets, was not met. The remaining data with a BP flag 3 corresponds to 2% of the silicate, 7% of the nitrate, and up to 8% of the phosphate samples. These flags are linked to the preservation technique applied (poisoned instead of frozen for nitrate and phosphate) which particularly explains the lower fraction of silicate samples that do not fulfill the "required" criteria (DYFAMED, OWSM). Note that internal analyses at DYFAMED resulted in favoring poisoning nutrients for conservation over storing them frozen, and that DYFAMED reversed back to the former method in 2012, as reflected in the large percentage of BP flags 3. However, such insights were not integrated into this assessment and underpin the need for regular workshops discussing and updating BP recommendations for ship-based time-series. In this context, we want to mention the recently started Euro GO-SHIP project (https://eurogo-ship.eu/), and in particular the related comparability assessment of different nutrient measurement protocols.

Nitrite and ammonium samples show slightly different patterns because the number of measured samples deviates from the above-described nutrients, i.e. the influence of the ALOHA nutrient samples is smaller.

Differences in the type of autoanalyzer (rapid flow analyzer or continuous segmented flow), storage duration and temperature, defrost procedure, carrier solution ("in-house" artificial seawater that resembles the nutrient concentrations of the region, "in-house" low nutrient seawater or commercially available OSIL standard), reference material (WAKO, OSIL, KANSO) and sample filtering were not considered in the evaluation. Such differences can also occur in time within a time-series program, as shown for nitrate in Fig. 4. Note that the dependency of the CVOO time-series on research vessels of opportunity results in multiple small methodological changes – e.g., instrument, sample volume, and whether the sample is analyzed at sea or stored frozen.

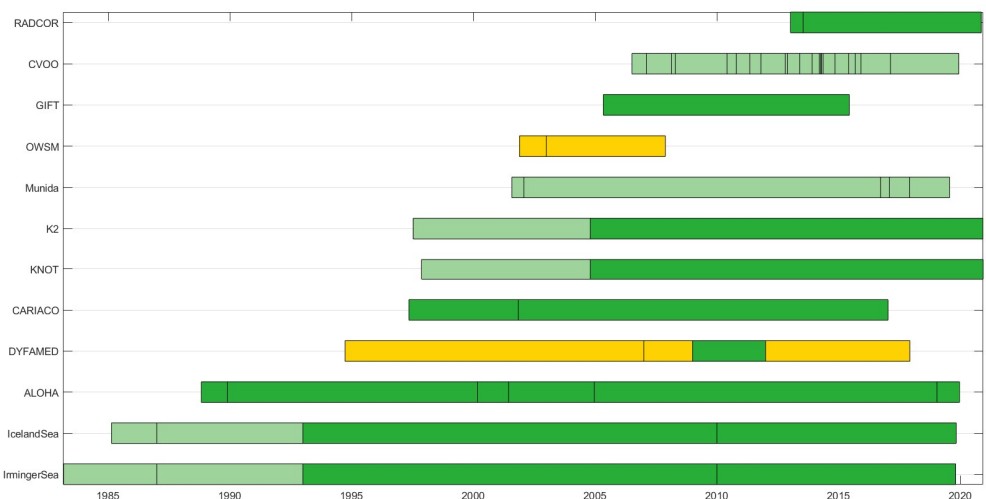

**Figure 4:** Time-dependency of assigned BP flag of each time-series program exemplarily shown for nitrate. Vertical black lines indicate method changes, captured and non-captured (e.g. instrument change) by the BP flags. The color scheme used, is identical to Fig. 3. Note that DYFAMED changed back to poisoning the samples for conservation based on internal analyses of conservation methods.

### 4.1.4. Dissolved Inorganic Carbon

For DIC, 88% of the samples were measured according to all BPs. DYFAMED is the only time-series program that measures DIC potentiometrically in a closed-cell. Even though DYFAMED made use of Dickson's CRMs since 1999, closed-cell potentiometric measurements of DIC alone have an offset (1-2% lower) (Bradshaw et al., 1981 and Millero et al., 1993), resulting in a BP flag 2. The remaining samples that do not meet the desired BP
requirements are pre-1991 samples from ALOHA, IRM-TS, and IC-TS, for which certified reference material was unavailable, also resulting in a BP flag 2.

Differences in sample storage duration and coulometer calibration methods (gas loop calibration or sodium carbonate solutions) were not considered in the evaluation. Very few samples for DIC are taken on the RADCOR cruises.


### 4.1.5. Total Alkalinity

Total alkalinity is one of the few variables measured by all participating time-series programs. 87% of the samples met all BP requirements; 10% the "required" requirements only; and 3% did not meet the required BP. The latter correspond to cruises for which metadata on total alkalinity are not present (ALOHA cruises 1-22) and to cruises
where total alkalinity was measured using a single-point titration (only few cruises at DYFAMED, K2 and KNOT) (Fig. 5). The BP flags of 2 are either linked to i) missing information on the indicator, cell type, and/or curve fitting method used, or ii) non-application of certified reference materials. Differences in storage duration, cell type, end-point, and curve fitting method (least-square or modified Gran functions) were not considered in the evaluation.

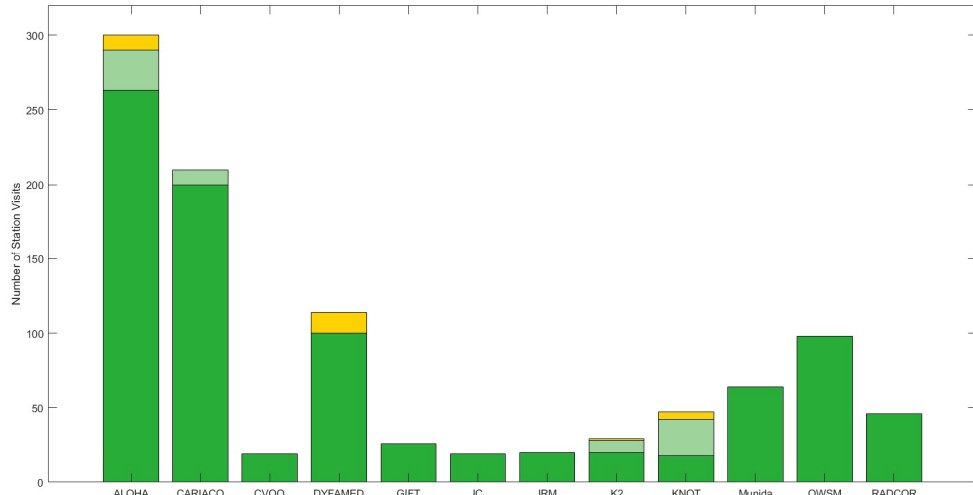


**Figure 5:** Assigned BP flags per station exemplarily shown for total alkalinity. Flags have been assigned on a cruise-per-cruise basis, i.e. per station visit. The color scheme used, is identical to Fig. 3.

### 4.1.6. pH

Even though most programs which analyze pH follow the methodology of Clayton and Byrne (1993), pH has the lowest number of programs with methods meeting all the BP requirements. CARIACO's protocol is the only one which meets all pH BP requirements, as reflected in the overall percentage of samples with a BP flag of 1 being only 39%. ALOHA, GIFT, and RADCOR reported pH on the total scale at 25°C and 0 dbar and analyzed pH using unpurified m-cresol purple. But none of these programs corrected for the impurities of the dye (54% of the
samples), thereby not meeting the BP flag 1 criteria. A few cruises of DYFAMED, K2 and KNOT measured pH, but pH was measured potentiometrically (less stable and accurate, Lorenzoni and Benway, 2013).

Differences in the storage duration, and more importantly whether an additional correction for pK* of the indicator dye m-cresol purple was applied (suggested by DelValls and Dickson, 1998), were not part of the BP flag evaluation. The latter correction has been applied by GIFT and CARIACO.




### 4.1.7. Partial Pressure of CO₂

The only two time-series programs that measure partial pressure of $CO_2$ ($pCO_2$) are the IRM-TS and IC-TS, both being measured by the same personnel using identical protocols. The presently applied protocol meets all BP requirements. Before mid-1993, the samples (3% of the total) were not poisoned for storage, but instead
equilibrated gas was isolated and sealed in a 300 mL glass flask. Further temporal changes of the methodology are explained in Olafsson et al. (2010).

### 4.1.8. Particulate Matter

Particulate matter concentrations are only measured at ALOHA, CARIACO, and CVOO. ALOHA and CARIACO
meet all BP requirements for particulate organic carbon and nitrogen, whereas CVOO (<1% of all samples) is missing information on standards used. ALOHA's particulate phosphorus measurements (75% of all samples) also meet all BP requirements, but CARIACO's metadata do not include details on the filter used for these measurements. CVOO also lacks detailed metadata for particulate phosphorus.

Differences in storage duration, and more importantly, filter sizes and types, heating temperature and duration,
and leaching time were not part of the evaluation. According to ALOHA's protocols, differences in the latter resulted in large variations of the measured particulate phosphorus content. ALOHA particulate organic phosphorus samples pre-2012 are biased low. Also, note that ALOHA particulate matter includes inorganic components.

### 4.1.9. Dissolved Organic Carbon (DOC)

ALOHA, CARIACO, GIFT, and K2 have measured DOC, and the samples of CARIACO, GIFT and K2 have been filtered. Thus, 33% of the DOC samples have a BP flag 1, and all samples from ALOHA (67%) received a BP flag 2.

### 4.2. Minimum Variability

The layers with the lowest oxygen variability (0.7% - 3.4%) are all located below 1000 dbar and represent the bottom layer in the cases of ALOHA, DYFAMED, and the IRM-TS (Table 4). For CVOO, IC-TS, K2, and OWSM, the determined layers are "near-bottom" to intermediate layers, probably reflecting that oxygen concentrations at the bottom are more prone to boundary layer effects in these regions. At KNOT, we can link this layer to the continual influx of NPDW.
Salinity shows the lowest variability for all time-series stations ranging from 0.003% - 0.086%. The higher values indicate that natural variability likely had a strong influence on the calculated numbers. Silicate is generally the nutrient with the highest variability within and across the time-series programs, with the IRM-TS experiencing the highest variability (6.7%). Such a high coefficient of variation cannot solely be linked to large uncertainties in the measurements (silicate accuracies ($V_{crm}$) at the IRM-TS are around 3.5%). Hence, natural variabilities of the
nutrients are very high in this region in the determined layer, which also corresponds with the upper end of the salinity variability. Nonetheless, silicate, having the highest of all nutrient variabilities, fits well to the assigned accuracy values and also to previous findings of rather high uncertainties in silicate concentrations (e.g., inter-laboratory studies described in Bakker et al., 2016) and experiences from the GLODAP quality control (Olsen et al., 2016). The coefficients of variation of DIC and total alkalinity are below 0.5% for all time-series stations with
a maximum of 0.4% (around 9 µmol kg⁻¹) at DYFAMED and a minimum of 0.1% (around 2 µmol kg⁻¹) at ALOHA, K2, KNOT, and CVOO. The latter are within the provided accuracy estimates and indicate very constant DIC and total alkalinity data quality. Minimum pH variability could only be calculated for ALOHA (0.04%), which is in the range of the provided pH precision values at ALOHA. DOC variabilities could be calculated for ALOHA and K2. For the former, it is 8.5% and thus around twice as large as given accuracy and precision values. For the latter,
it is 1.7% and fits very well with the provided precision values. For the IRM-TS and IC-TS $pCO_2$ data, the determined coefficients of variation are two to three times as large as the stated precision (Olafsson et al., 2010), which again can be linked to the rather high natural variability of all variables at these stations. No minimum consistencies could be calculated for particulate matter.

The obtained minimum variabilities can in some cases (e.g., ALOHA) be interpreted as an inter-consistency
determination of the measurement quality. In these cases, low variability indicates a consistent level of data quality throughout the measurement period. A high variability then indicates a variable level of data quality. Here, the determined layers can also be used to detect suspicious samples. However, some sites (e.g., IRM-TS) are characterized by large natural variability on all depth surfaces (on several timescales), likely accompanied and recognizable by high salinity variability. For these stations, the high variability estimates should not be confused
with a high variability in measurement quality.



**Table 4**: Minimum variability expressed as the coefficient of variation (%). The corresponding layer depth of the layer with the least oxygen variability (+/- 100 dbar) on which the variabilities have been calculated, is shown, too. The variable abbreviations are the same as in Table 1. The Rhombus denotes that CTD values have been used for the calculation.

| | Layer | S | $O_2$ | $NO_3$ | $PO_4$ | $SiOH_4$ | DIC | TA | pH | $pCO_2$ | DOC |
|---|---|---|---|---|---|---|---|---|---|---|---|
| **ALOHA** | 4400 dbar | 0.005 | 0.7 | 0.7 | 0.8 | 0.8 | 0.1 | 0.2 | 0.04 | NA | 8.5 |
| **CARIACO** | NA | NA | NA | NA | NA | NA | NA | NA | NA | NA | NA |
| **CVOO** | 3000 dbar | 0.008 | 0.8 | 2.2 | 2.8 | 2.5 | 0.2 | 0.1 | NA | NA | NA |
| **DYFAMED** | 2400 dbar | 0.033# | 1.8 | 3.3 | 4.3 | 5.1 | 0.4 | 0.3 | NA | NA | NA |
| **GIFT** | NA | NA | NA | NA | NA | NA | NA | NA | NA | NA | NA |
| **IcelandSea** | 1200 dbar | 0.017 | 1.4 | 3.4 | 4.9 | 5.1 | 0.2 | 0.3 | NA | 2 | NA |
| **IrmingerSea** | 1000 dbar | 0.086 | 3.4 | 4.2 | 5.3 | 6.7 | 0.3 | 0.4 | NA | 3 | NA |
| **K2** | 5000 dbar | 0.003 | 0.6 | 0.4 | 0.5 | 1.5 | 0.1 | 0.1 | NA | NA | 1.7 |
| **KNOT** | 3800 dbar | 0.011 | 0.7 | 0.4 | 0.6 | 1.0 | 0.1 | 0.1 | NA | NA | NA |
| **Munida** | NA | NA | NA | NA | NA | NA | NA | NA | NA | NA | NA |
| **OWSM** | 1200 dbar | 0.009# | 0.7 | 2.5 | 4.2 | 6.5 | 0.2 | 0.3 | NA | NA | NA |
| **RADCOR** | NA | NA | NA | NA | NA | NA | NA | NA | NA | NA | NA |




### 4.3. Comparison to GLODAP

The relaxation of the crossover analysis (Sect. 3.4) enabled the determination of offsets between GLODAP and time-series stations of ALOHA, CVOO, IC-TS, IRM-TS, KNOT, K2, and OWSM (Table 5). Generally, the analysis indicates a very good fit between the SPOTS pilot and GLODAP at these sites. Significant offsets suggest the potential for bias in either the SPOTS pilot or GLODAP, but further analysis of both products is required to assess the source of bias. In the following, the results are presented for each time-series program individually.

**Table 5**: Mean offsets (rounded) of the SPOTS pilot against GLODAP core variables. The first number in parentheses shows the number of cruises from the time-series program compared to GLODAP. The second number in the parentheses shows the total number of cruises from GLODAP to which the time-series cruises are compared. The variable abbreviations are the same as in Table 1. The Asterix denotes whenever the crossover analyses have been performed on pressure surfaces. The Rhombus denotes that CTD values have been used for the calculation. NPDW stands for North Pacific Deep Water.

| | S | O$_2$ | NO$_3$ | PO$_4$ | SiOH$_4$ | DIC | TA | pH | Layer |
|---|---|---|---|---|---|---|---|---|---|
| **ALOHA** | 0.0019 (3;1) | 0% (3;1) | NA | -2% (3;1) | -1% (3;1) | NA | NA | NA | 2000 dbar – bottom |
| **CARIACO** | NA | NA | NA | NA | NA | NA | NA | NA | 500 dbar – bottom |
| **CVOO** | 0.0003 (6;9) | 0% (8;9) | 0% (6;9) | 1% (6;9) | 1% (4;5) | 0 µmol kg$^{-1}$ (2;4) | 1 µmol kg$^{-1}$ (2;4) | NA | 1500 dbar – bottom |
| **DYFAMED** | NA | NA | NA | NA | NA | NA | NA | NA | NA |
| **GIFT** | NA | NA | NA | NA | NA | NA | NA | NA | NA |
| **IcelandSea*** | -0.0006 (5;4) | 1% (3) | -2% (4;3) | -6% (4;3) | -4% (4;3) | -2 µmol kg$^{-1}$ (2;2) | NA | NA | 1000 dbar – bottom |
| **IrmingerSea*** | 0.0068 (5;6) | -3% (1;3) | -4% (1;2) | -1% (1;2) | 6% (1;1) | NA | NA | NA | 500 dbar – bottom |
| **KNOT** | 0.0002 (28;41) | 0% (28;37) | 0% (29;39) | 0% (29;39) | -1% (27;35) | -2 µmol kg$^{-1}$ (28;35) | -5 µmol kg$^{-1}$ (30;37) | -0.005 (1;1) | NPDW |
| **K2** | 0.0004 (15;17) | 0% (16;18) | 0% (15;17) | 0% (16;18) | -1% (15;17) | 0 µmol kg$^{-1}$ (16;18) | -3 µmol kg$^{-1}$ (14;16) | -0.005 (1;1) | NPDW |
| **Munida** | NA | NA | NA | NA | NA | NA | NA | NA | NA |
| **OWSM*** | 0.0023$^{\#}$ (2;4) | 0% (1;1) | -3% (1;1) | -3% (1;1) | -1% (1;1) | 6 µmol kg$^{-1}$ (2;4) | NA | NA | 1000 dbar – bottom |
| **RADCOR** | NA | NA | NA | NA | NA | NA | NA | NA | NA |

### 4.3.1. ALOHA

For ALOHA all calculated crossover offsets fall within the provided GLODAP consistencies (Lauvset et al., 2021), indicating a good fit between the two products. There are no crossover cruises for nitrate and carbon variables. Further, only three ALOHA cruises (HOT174 - HOT176) are compared against only one GLODAP cruise (49NZ20051031), as these are the only crossover pairs that meet the crossover criteria. Note that 49NZ20051031 has passed the full 2$^{nd}$ QC of GLODAP and that the individual crossover pairs offsets are similar. Nonetheless, the small amount of underlying data strongly reduces the confidence in the results.

### 4.3.2. CVOO

Crossover offsets could be calculated for all GLODAP core variables which were measured at CVOO. All analyzed variables fall clearly within the provided GLODAP consistencies, indicating a good fit between the two products at CVOO. The results are robust, given the number of CVOO cruises compared to GLODAP. Further, there is very good agreement between the individual crossovers, i.e. low standard deviations of the individual offset between one cruise and GLODAP, and consistency among all CVOO cruise offsets with no large outliers. Data from a few cruises are present in both products.



### 4.3.3. Iceland Sea

The crossover offsets of the IC-TS of salinity, oxygen, nitrate, and DIC against GLODAP are within the consistency limits of GLODAP, i.e. no significant offset is remarkable between the two products. For nitrate, the variability between the individual offsets is large, which reduces confidence in the analysis. For phosphate, the SPOTS pilot has 6% lower concentrations than GLODAP based upon four cruises from the IC-TS (B17-94, B9-96, B12-96, and B5-2002) and three GLODAP cruises (58JH19941028, 58JH19961030 and 316N20020530), which all passed GLODAP's 2nd QC. This large offset mainly originates from the 2002 cruise, while cruises from 1996 indicate a good fit. The same cruises show a -4% offset for silicate, and the underlying data show a similar pattern. However, the relatively large minimum variability of salinity (Sect. 4.2) demonstrates that the Iceland Sea is a dynamically active region with deep open ocean convection and complex seasonally varying currents; this high natural variability reduces confidence in the crossover analysis for the Iceland Sea region.

### 4.3.4. Irminger Sea

All crossover offsets of the IRM-TS against GLODAP are above GLODAP's consistency limits except for phosphate. However, given i) that the minimum depth had to be set to only 500 m in a deep water formation area and ii) the relatively large minimum variability of salinity (Sect. 4.2), the larger offsets were expected and are likely attributable to the inherent natural variability of this region. Further, the relatively small number of crossovers does not allow for a more in-depth investigation of the offsets.

### 4.3.5. KNOT

Crossover offsets could be calculated for all GLODAP core variables. The calculations were performed on the NPDW, which has a residence time of about 500 years (Stuiver et al., 1983). Following the definition from Wakita et al. (2010), we used 27.69 $\sigma$ (around 2000 dbar) and 27.77 $\sigma$ (around 3500 dbar) as limits. All of the so-calculated offsets of KNOT against GLODAP are clearly within the consistency limits except for total alkalinity (-5 µmol kg$^{-1}$). Confidence in the analysis is provided through a large number of crossover cruises and consistency of calculated offsets. Data from a few cruises are present in both products.

### 4.3.6. K2

Crossover offsets could be calculated for all GLODAP core variables. The calculations were again performed on the NPDW using the identical limits as those of KNOT. All of the so-calculated offsets of K2 against GLODAP are clearly within the consistency limits. Confidence in the analysis is provided through a large number of crossover cruises and consistency of calculated offsets, as exemplarily shown for nitrate (Fig. 6). Data from a few cruises are present in both products.

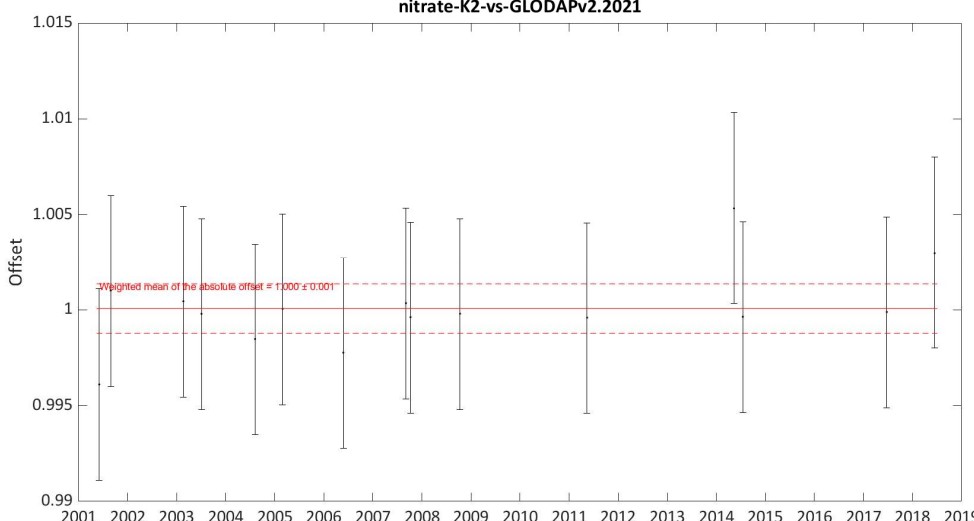

**Figure 6**: Total weighted offset of the SPOTS pilot nitrate data against GLODAPv2.2021 at station K2 in the North Pacific Deep Water (NPDW) layer. The total weighted offset is multiplicative and illustrated by the red line. The dashed red lines are





the corresponding standard deviation. The black dots display the weighted offsets of individual K2 cruises against GLODAP cruises with the corresponding error bars displaying their standard deviation. If the calculated standard deviation of the individual cruises is lower than GLODAP's nitrate consistency limit (2%) it is set to the latter. The summary figure indicates a very good fit between the SPOTS pilot product and GLODAP at the K2 station for nitrate with a total weighted offset of 0.0%.

### 4.3.7. OWSM

Crossover offsets at OWSM indicate slight mismatches between the nitrate, phosphate, and DIC concentrations of the SPOTS pilot vs. GLODAP. The total weighted mean offsets are -3%, -3%, and 6 µmol kg$^{-1}$, respectively. The former two offsets are only based upon a comparison between the OWSM cruise from 20020415 (no CRUISE ID present) and 316N20020530. Three more recent OWSM cruises from 2019 are additionally checked against 720 58JH20190515. Both GLODAP cruises passed GLODAP's 2$^{nd}$ QC. However, the DIC offsets are very dependent on the crossover pair and the final offset should be treated with caution. The small number of crossovers does not allow for a more in-depth investigation of the relatively small offsets.

## 5. Product File Description

The product file variable names are described in Table S8. Each fixed-location time-series station is identified by the entry under "TimeSeriesSite", and individual cruises are identified by "CRUISE". Station, cast, and bottle numbers are linked to the original cruise campaign numbering (if provided). In some cases, station number duplicates within the same time-series program exist as the data originates from different research vessels of opportunity (Table 1). Nitrate values can contain nitrite concentrations (Table S4). Similarly, ALOHA's particulate organic matter includes particulate inorganic components. Since all pH values were reported on the total scale at 25°C, no additional pH temperature entry is provided. Conversely, for $pCO_2$ corresponding temperature measurements are given. In addition to the WOCE flags, each bottle variable is further accompanied by the assigned BP Flag (Sect. 4.1) and by the provided precision and accuracy estimates (Sect. 3.4). The last column lists the digital object identifier (DOI) of the original dataset. All missing entries are indicated by -999.

A total of 108,332 water samples are included in the product. Bottle salinity with 75,654 measurements is the variable with the most abundant data (Table 6). The number of bottle salinity samples is about twice the number of bottle oxygen and nutrient (excluding ammonium and nitrite) samples and almost five times the number of included DIC and total alkalinity samples. pH and nitrite have around 10,000 samples and the product includes between 4,900 and 7,600 samples of particulate matter, DOC, and ammonium. With 1,898 samples from the IRM-TS and the IC-TS, $pCO_2$ is the variable with the fewest measurements. Silicate, nitrate, and total alkalinity are the only variables measured at all sites. Around 56% of all bottle data values originate from ALOHA (Table 6) and 14% from CARIACO. The remaining 25% are distributed rather equally across the different programs. ALOHA's large percentage can be explained by measurements at ALOHA i) having taken place consistently on a monthly basis for ≥30 years; ii) including up to 30 hydrocasts per station visit; and iii) including all but two of the product's bottle variables. The dominance of ALOHA's measurements is most pronounced for salinity, particulate phosphate (inorganic and organic), and DOC (around 70% - 80% of the samples are measured at ALOHA). For oxygen and nutrients, ALOHA's samples represent around 52% of all samples, and for the inorganic carbon variables (DIC, total alkalinity, and pH) between 32% - 42%.

**Table 6**: Summary statistics showing the total number of samples per variable included in the SPOTS pilot of each time-series site. Percentages in brackets show fractions in comparison to the total number per variable except for the last column. Percentages are rounded; thus, the sum is not always equal to exactly 100%. Variable abbreviations are identical to Table 1.

| | S | O₂ | NO₃ | NO₂ | PO₄ | SiOH₄ | NH₄ | DIC | TA | pH | pCO₂ | POC | PON | POP | DOC | Total |
|---|---|---|---|---|---|---|---|---|---|---|---|---|---|---|---|---|
| **ALOHA** | 63334 (84%) | 21937 (57%) | 18130 (52%) | 750 (6%) | 17648 (53%) | 17656 (52%) | 0 | 5911 (35%) | 5780 (32%) | 4124 (42%) | 0 | 3659 (48%) | 3637 (49%) | 3675 (75%) | 4778 (67%) | 171019 (56%) |
| **CARIACO** | 4026 (5%) | 3528 (9%) | 3705 (11%) | 3768 (32%) | 3724 (11%) | 3691 (11%) | 3680 (69%) | 0 | 3687 (21%) | 3760 (39%) | 0 | 3870 (51%) | 3804 (51%) | 1221 (25%) | 975 (14%) | 43439 (14%) |
| **CVOO** | 345 (<1%) | 534 (1%) | 451 (1%) | 507 (4%) | 451 (1%) | 411 (1%) | 73 (1%) | 346 (2%) | 304 (2%) | 0 | 0 | 39 (1%) | 39 (1%) | 24 (<1%) | 0 | 3524 (1%) |
| **DYFAMED** | 0 | 2328 (6%) | 1525 (4%) | 1670 (14%) | 1611 (5%) | 1482 (4%) | 0 | 1086 (6%) | 1114 (6%) | 56 (1%) | 0 | 0 | 0 | 0 | 0 | 10872 (4%) |
| **GIFT** | 0 | 480 (1%) | 479 (1%) | 0 | 0 | 477 (1%) | 0 | 0 | 470 (3%) | 463 (5%) | 0 | 0 | 0 | 0 | 199 (3%) | 2568 (1%) |
| **IcelandSea** | 2214 (3%) | 2111 (5%) | 2070 (6%) | 0 | 2087 (6%) | 2101 (6%) | 0 | 1824 (11%) | 280 (2%) | 0 | 1101 (58%) | 0 | 0 | 0 | 0 | 13788 (4%) |
| **IrmingerSea** | 1901 (3%) | 1792 (5%) | 1774 (5%) | 0 | 1767 (5%) | 1784 (5%) | 0 | 1477 (9%) | 209 (1%) | 0 | 797 (42%) | 0 | 0 | 0 | 0 | 11501 (4%) |
| **K2** | 1921 (3%) | 1904 (5%) | 1996 (6%) | 1997 (17%) | 1994 (6%) | 1983 (6%) | 1188 (22%) | 1897 (11%) | 1805 (10%) | 509 (5%) | 0 | 0 | 0 | 0 | 1129 (16%) | 18323 (6%) |
| **KNOT** | 1864 (2%) | 1997 (5%) | 1859 (5%) | 1893 (16%) | 1851 (6%) | 1862 (5%) | 376 (7%) | 1821 (11%) | 1802 (10%) | 174 (2%) | | 0 | 0 | 0 | 0 | 15445 (5%) |
| **Munida** | 0 | 0 | 285 (1%) | 0 | 285 (1%) | 280 (1%) | 0 | 220 (1%) | 298 (2%) | 0 | 0 | 0 | 0 | 0 | 0 | 1368 (<1%) |
| **OWSM** | 49 (<1%) | 905 (2%) | 1004 (3%) | 0 | 911 (3%) | 1004 (3%) | 0 | 2053 (12%) | 1320 (7%) | 0 | 0 | 0 | 0 | 0 | 0 | 7246 (2%) |
| **RADCOR** | 0 | 1215 (3%) | 1270 (4%) | 1279 (11%) | 1268 (4%) | 1284 (4%) | 0 | 190 (1%) | 739 (4%) | 678 (7%) | 0 | 0 | 0 | 0 | 0 | 7923 (3%) |
| **Total** | 75654 | 38731 | 34548 | 11810 | 33597 | 34015 | 5317 | 16825 | 17808 | 9764 | 1898 | 7568 | 7480 | 4920 | 7081 | 307016 |



## 6. Stakeholders

The main stakeholder groups of SPOTS are the data providers on the upstream-end, i.e. the individual time-series programs (Sect. 2), and users of time-series data on the downstream-end. Regarding the latter, the SPOTS pilot is intended to be applied in different ocean BGC fields: evaluations of ocean BGC, neural networks such as CANYON-B (Bittig et al., 2018), CANYON-MED (Fourrier et al. 2020), or ESPER (Carter et al., 2021), regional ocean BGC models, (e.g., models participating in RECCAP such as Ishii et al., 2015), 1D model applications (e.g., Mamnun et al., 2022 using REcoM2), global ocean BGC models participating in model intercomparison projects (e.g., Coupled Model Intercomparison Project - Orr et al., 2016); evaluations of autonomous BGC observing networks such as BGC Argo (Bittig et al., 2019); global scientific assessments such as the Global Carbon Budget (Friedlingstein et al., 2022); or multi time-series studies and analyses (e.g., Bates et al., 2014; O'Brien et al., 2017). These time-series can also contribute ocean carbonate chemistry data to the United Nations Sustainable Development Goals, especially target 14.3 to minimize and address the impacts of ocean acidification.

### 6.1. Benefits

The main goal of SPOTS is that both stakeholder groups benefit from the product. Through a use-case, the benefits for the users are implicitly demonstrated in Sect. 6.2.

On the upstream end, data providers benefit from the product in several ways. First of all, the product increases the impact of individual ship-based time-series programs. For smaller and less well-known time-series programs, the impact is particularly improved by increasing their visibility and discoverability. Here, two "pull factors" contribute: i) the popularity and success of the included larger time-series programs and ii) being exposed on the Ocean Data and Information System (ODIS) catalog (https://book.oceaninfohub.org) in a schema.org-friendly way (Sect. 7.2). The larger sites also benefit from the latter, but the impact of larger time-series programs is in particular increased through enhanced usability of their data. Here, the proverb "the whole is greater than the sum of its parts" perfectly describes the benefits of SPOTS. The envisioned (non-exhaustive) list of users underscores the idea that consistent and inter-comparable data from multiple time-series programs (i.e. the "whole") leads to an extended range of applications relative to those of a single time-series program. The data being automatically uploaded to ERDDAP, which increases the accessibility, interoperability, and machine-readability (Sect. 7.2), also becomes important in broadening users and applications of data from these time-series programs.

Further, participating time-series programs benefit from optional data management support for formatting, QC, and data archival. This support aims at reducing the data management workload of individual programs and being directly ascribed to the FAIR data practices. Regarding guidelines and BPs, the participating time-series programs also benefit from the product fostering collaborations across several programs, which is especially relevant for emerging time-series programs.

Ship-based time-series programs represent one of our most powerful tools for monitoring marine ecosystem changes. The product contributes to the development of a sustained, globally distributed network of time-series observatories that sample a core set of biogeochemical and ecological variables guided by common best practices (methodological, FAIR data, etc.). These are required attributes of a GOOS observing network, and achieving this status would ultimately help position ship-based time-series programs for expansion under the United Nations Decade of Ocean Science umbrella. In addition, the product links individual time-series efforts to larger policy directives such as the Marine Strategy Directive Framework in Europe with respect to e.g., ocean monitoring indicators.

### 6.2. Use-Case

As an example to demonstrate both the utility and potential misuses of the SPOTS pilot, we applied the recently developed Trends of Ocean Acidification Time Series software (TOATS, https://github.com/NOAA-PMEL/TOATS) to the mixed layer total alkalinity data included in the product (Fig. 7). The TOATS software is a supplement to the recently published best practices for assessing trends of ocean acidification time-series and provides a python based Jupyter Notebook to compare trends across different (BGC) time-series data sets (Sutton et al., 2022). It was developed based on several published trend analysis techniques to standardize estimating and reporting trends from ocean carbon time-series data sets. Following a strict sequence of approaches[8], TOATS

---

[8] 1. assess data gaps in the time-series; 2. remove periodic signals (i.e. normally occurring variations due to predictable cycles) from the time-series;3. assess a linear fit to the data with the periodic signal(s) removed; 4. estimate whether a statistically significant trend can be detected from the time-series; 5. consider uncertainty in the measurements and reported trends; and 6. present trend analysis results in the context of natural variability and uncertainty.



estimates i) the linear trend, ii) its uncertainties, and iii) the trend detection time of the assessed time-series data. The latter indicates the minimum observational period needed to statistically distinguish between natural variability (noise) and anthropogenic forcing. This method requires time-series with sub-seasonal sampling frequency to constrain seasonal variability of surface ocean carbonate chemistry; however, for the purpose of this

example, we assessed all time-series programs rather than restricting the assessment to time-series datasets with regular monthly measurements. The only non-trivial calculation step we applied before running TOATS was to calculate the surface mixed layer depth for each cruise (defined using a 0.3 potential density anomaly criteria following de Boyer Montégut et al. (2004)) and to average total alkalinity concentrations within the estimated mixed layers. The results of our use-case (Fig. 7) show trends in alkalinity for all time-series (seven of them with

significant trends).

The ease of use in applying TOATS to multiple time-series demonstrates the main benefits and potential misuse of the SPOTS pilot at the same time. Concerning the benefits, the combination of the SPOTS pilot and TOATS enables any user to perform joint time-series studies that follow published BPs without requiring any in-depth programming knowledge. The need to, a priori, know about existing time-series program data and to subsequently

mine, format, and QC the data, becomes redundant for all time-series datasets included in SPOTS. The required input format of TOATS is also readily available by accessing the time-series product data through ERDDAP (Sect. 7.2). Further, detailed information on methods and their changes over time will become even more accessible once the ODIS user interface is online. This will enable a sophisticated information-driven data selection of (subsets of) time-series data to analyze the effects of method changes on detected trends without having to study multiple

cruise reports. A similar advantage is provided through the possibility of selecting subsets of data based on the assigned BP flags (Sect. 4.1). Lastly, the estimates of precision and accuracy included in the SPOTS pilot (Sect. 3.4) additionally enable confident uncertainty estimations of the trend analyses (uncertainties of the observations being a mandatory input in TOATS).

Regarding the potential misuse of the SPOTS pilot, caution must be applied in interpreting the results, particularly

because the use-case analysis includes values accompanied by BP flags 2 and 3. Simply assuming that the determined trends (Fig. 7) are valid and interpreting differences across time-series programs could lead to false conclusions. Robust trend analysis also requires the user to acknowledge the impact of large data gaps in time-series that inhibit the ability to constrain seasonal variability in many of the included datasets (e.g. CVOO), and make it impossible to remove periodic signals with confidence (second step of TOATS trend analysis). Following

TOATS guidelines, we recommend applying TOATS to surface ocean biogeochemical data with at least regular seasonal measurements or to restrict the trend analysis to specific seasons. Increasing the number of samples using additional interpolation and computational techniques could relax this restriction (e.g., multivariate linear regression (MLR; Vance et al., 2022), but computations accompanied by large uncertainties might also harm the robustness of the trend analyses. Note that in the case of interpolating concentrations of single variables vertically,

we recommend using a quasi-Hermetian piecewise polynomial (Key et al., 2010). And if techniques to increase the data coverage involve using CO2SYS (van Heuven et al., 2011), we recommend using the carbonate dissociation constants of Lueker et al. (2000), the bisulfate dissociation constant of Dickson (1990), and the borate-to-salinity ratio of Uppström (1974).

Another large pitfall is neglecting the provided metadata and assuming that restricting the analyses to time-series

data with a BP flag 1 erases all artifacts in the trend analyses. Such a restriction would increase the robustness of the analysis, but unaccounted differences within the BP flag 1 (Sect. 4.1) would still bias the results. For example, ALOHA particulate phosphorus samples analyzed before 2012 are biased low but still fulfill all assessed BP requirements (Sect. 4.1). Similarly, some standardizations of the product resulted in the neglect of valuable time-series details (e.g., information on ventilation events provided through the unique QC flags of CARIACO (Sect. 3.2)).

We included all information in the additional metadata, made it easily accessible, and encourage users consult it, particularly to check for any correlations of the trend analyses to method changes (Table S6) and/or specific time-series events.

Even though this example highlights a multiple time-series study use-case, it depicts the benefits and especially

the potential misuses for other applications of the SPOTS pilot. If the limitations of the product (e.g., data gaps and varying baselines) are acknowledged, quality descriptors are utilized, and the data are used in conjunction with the supporting metadata, multiple applications can benefit from this time-series product.

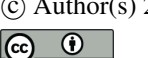

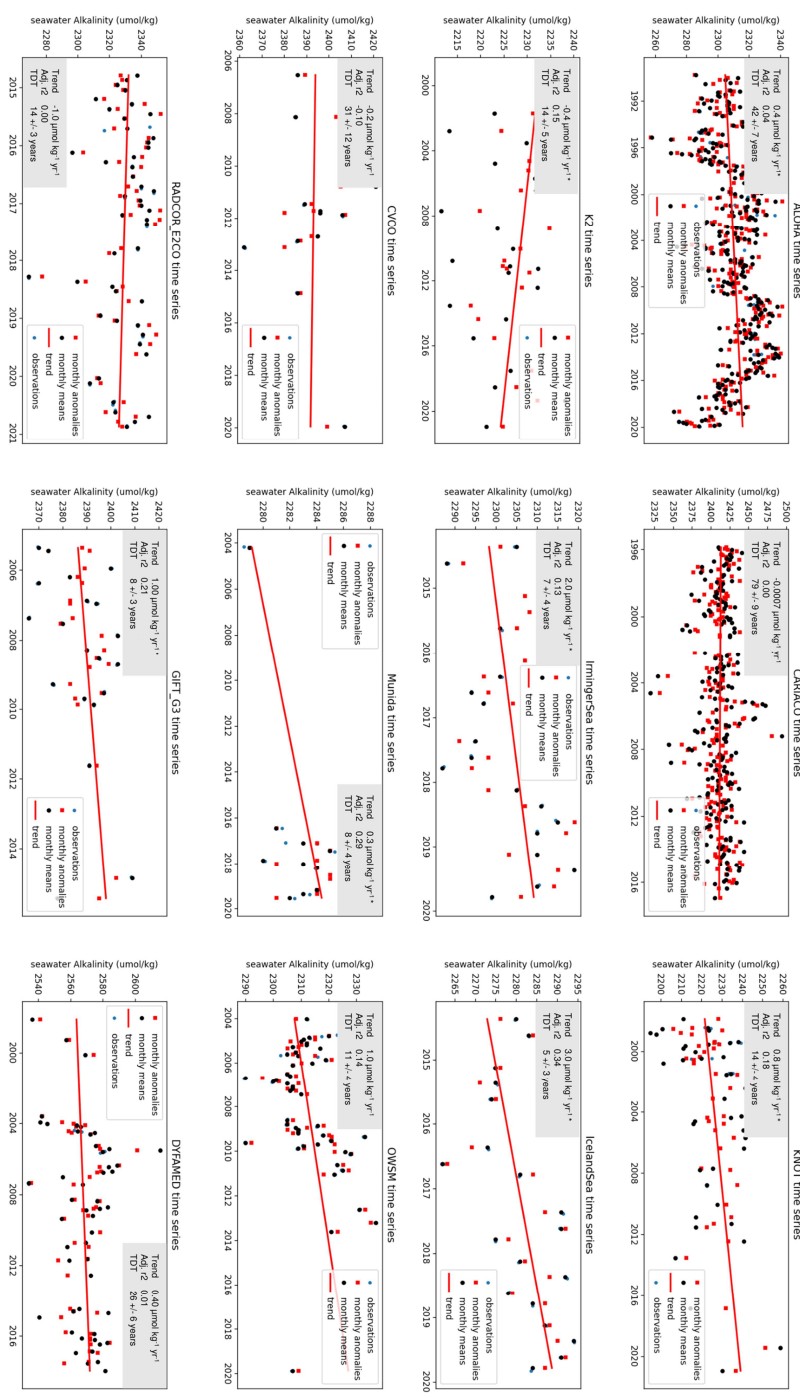

**Figure 7**: Trend analysis of total alkalinity in the mixed layer using TOATS. Data symbols show the original time-series observations (blue circles), the time series of monthly means (black circles), the de-seasoned monthly means (red squares), and the trend of the de-seasoned monthly means (red line) (From Sutton et al., 2022). The monthly anomalies (red squares) that are used for the trend analyses are shown as de-seasoned monthly means. The grey boxes include the yearly trend, adjusted R2 and the minimum trend detection time (TDT). An Asterix next to the yearly trend number indicates that the result is significant (two-sided t-test p-value < 0.05). Note that x- and y-axis are not in synch among the different time series subplots.





**6.3. Recommended Standard Operating Procedures (SOPs) for Ship-Based Time-Series Programs**

The process of generating the SPOTS pilot resulted in the formulation and recommendation of SOPs regarding metadata documentation, internal QC, and uncertainty estimation. These are directed at the data providers, i.e. those who help run the ship-based time-series programs. The proposed SOPs are briefly presented here, the full guidelines can be accessed at https://www2.whoi.edu/site/mets-rcn/.

1. **Metadata documentation**: The first SOP is the recommended metadata template (Table S2), which provides a structure for time-series programs to uniformly document the applied methodologies, thereby ensuring that relevant information, including differences between individual cruises, is recorded. It should be filled out for each cruise individually. The metadata enables detailed method comparisons of ship-based BGC EOV data such as the BP assessment of the participating sites of the data product. We recommend that the metadata template be updated as the community re-determines, expands, and specifies the BPs for BGC EOV ship-based time-series data.

2. **Consistent QC routine:** The second SOP recommendation involves the use of a consistent routine to QC time-series data. The main goal is that scientists follow consistent criteria to flag single samples. Different characteristics of time-series programs - e.g., location (depth and seasonal influence), funding opportunities (duration and frequency of visits), and scientific goals (variables measured) - preclude a "one-size-fits-all" QC method. Thus, a decision tree approach guides the user in choosing the appropriate type of QC for their dataset. All suggested semi-automated checks make particular use of comparisons with historical time-series data. To evaluate the flagging results, the SOP is accompanied by a comparison to the well-established HOT QC results.

3. **Calculating uncertainty:** The third SOP has been developed by the Oslo and Paris Conventions Commission (OSPAR), Hazardous Substances & Eutrophication Committee (OSPAR, 2011) and was originally intended for assessments of contaminants in biota and sediment done in OSPAR areas. It can also be applied to BGC EOV ship-based data. It provides detailed recommendations for a consistent estimation of one total measure of uncertainty, including exact formulas that combine the information obtained through duplicate measurements (precision) and comparisons to reference material (accuracy).



## 7. Data Access and Availability

### 7.1. METS-RCN Website

All information regarding the SPOTS pilot and the collaborative NSF EarthCube funded Marine Ecological Time Series Research Coordination Network (METS-RCN) can be accessed at https://www2.whoi.edu/site/mets-rcn/. The SPOTS web page (https://www2.whoi.edu/site/mets-rcn/ts-data-product/) includes detailed information on the participating time-series programs, including:

- contact person(s)
- time-series website URL
- relevant data repositories
- cruise reports and papers
- detailed metadata on the BGC EOVs measured
- recommended SOPs (Sect. 6.3) and in-depth information on the assigned BP flags
- links to AtlantOS QC software and crossover toolbox used


The website also provides several options for users to download the SPOTS pilot (DOI: 10.26008/1912/bco-dmo.896862.1), including:

- Comma-separated value (CVS) format (directly from the website)
- Link to the BCO-DMO repository (https://www.bco-dmo.org/dataset/896862, Lange et al., 2023)
- GOOS-relevant ERDDAP server (https://data.pmel.noaa.gov/generic/erddap/tabledap/bgc_ts_product.html)

### 7.2. Environmental Research Division's Data Access Program (ERDDAP)

Providing the data through ERDDAP enables FAIR-compliant data access services and gives users significantly
enhanced capabilities rather than just downloading the dataset directly from the website. Optional constraints within the ERDDAP dataset enable downloading subsets of the dataset. The constraint options include amongst others variable-, station- and time selections. ERDDAP also enables downloading the dataset in several formats, such as tab-separated or netCDF. The latter format also entails additional metadata attributes, including alternative variable names (NERC P01 or following the recommendations from Liqing et al. (2022). On the ERDDAP server,
users find a link "Make a graph" (https://data.pmel.noaa.gov/generic/erddap/tabledap/bgc_ts_product.graph), which enables plotting the data using the web-based ERDDAP tool. In addition to giving the users more degrees of freedom, hosting the dataset on the ERDDAP server has two important benefits. First, the dataset is machine-readable, enabling an automated transfer to other repositories and higher-level infrastructures (e.g., SeaDataNet, Copernicus Marine Environment Monitoring Service). Second, ERDDAP data managers are working to provide
direct access to metadata information stored in the ODIS catalog, which, once achieved, will significantly improve metadata interoperability.

### 7.3. ODIS catalog

Through collaborating with ODIS, we developed two json-ld templates to publish time-series program metadata
in a schema.org-friendly way (inspired by Science on Schema; Shepherd et al., 2022) and to enable FAIR metadata. The first template (*EventSeries*) is designed to capture the general information about the time-series programs (e.g., location, time, principal investigators, funding, and related datasets). A "sub-events" section is used for more details about the individual cruise's location, time, personnel, and vessel. That section also includes details about the applied measurement methodologies for each cruise and provides links to cruise reports. The second json-ld
template (*Dataset*) is designed to describe the metadata of the related BGC discrete bottle datasets. Here, the included variables and in particular, the applied semantics of the dataset are described. By using and linking these templates for each of the participating sites, we could include the metadata of the time-series sites and related datasets in the ODIS catalog. Here, the time-series programs are exposed on the web and machine-readable (interoperable) access to the metadata is guaranteed. Presently, these json-ld files are hosted by the METS-RCN
GitHub repository (https://github.com/earthcube/METS-RCN). Eventually, the individual time-series program's data centers can host (and update) these files and assign unique identifiers. The metadata of the SPOTS pilot itself (*Dataset*) are also stored in the ODIS catalog, clearly linking all related metadata to the data synthesis product.



### 7.4. Fair Data Usage Agreement

While the SPOTS pilot is made available without any restrictions (Creative Commons Attribution 4.0.), users of the data should adhere to fair data use principles: For investigations that rely on data from a particular time-series program, principal investigators should be contacted to explore opportunities for collaboration and co-authorship and if there are any uncertainties regarding methodological details or interpretation of datasets. The original dataset DOI and any articles where the data are described should be cited. Contacting principal

investigators comes with the additional benefit of expert insight into the specific site under investigation. This paper should be cited in any scientific publications that result from the usage of the SPOTS pilot.

## 8. Conclusion

The SPOTS pilot synthesized data from 12 ship-based ocean time-series programs, each representative of a unique marine environment. Time-series data and metadata were compiled and assessed to provide an internally consistent data product. As a pilot study, for feasibility, the focus of this initial ship-based time-series data product was BGC EOV data, which served as a use-case for the METS RCN and provided a template for a sustained living data product for ocean time-series.

Through an external qualitative assessment of the applied methodologies, flags were assigned that reflect the degree to which BPs were followed, which determines the comparability of the data. The most recently applied methods typically met the required BPs, but measurements of oxygen and pH still show room for improvement. Though the methods are adequately documented by many time-series programs, several others need to document their methods more thoroughly. The assessment also revealed the need to determine the level of granularity of both required documentation and required BPs for fully comparable data. The importance of inter-laboratory studies (e.g., QUASIMEME) must be highlighted in this context. In addition to the included precision and accuracy estimates, quantitative assessments yielded additional indicators that describe the consistency within- and across the time-series programs. For time-series stations dominated by water masses that contribute negligible natural variability, the calculated minimum variabilities demonstrate a high continuity in measurement quality. Reasonable fits between GLODAP and the majority of the time-series programs further increase the confidence in the data quality.

By making BGC EOV datasets from multiple sources consistent and ready to use, the SPOTS pilot facilitates an improved understanding of the variability and trends of ocean biogeochemistry. It represents an important and necessary step forward in broadening our view of a changing ocean and maximizing the return on our continued investment in ship-based ocean time-series programs. It also enhances data readiness (Lindstrom et al., 2012) by implementing FAIR data practices for all included data. In particular, the implementation of ERDDAP and ODIS (Sect. 7.2) enables easy data integration into e.g., OceanOPS and Copernicus Marine Environment Monitoring Service. On a higher level, this effort facilitates the consolidation of the international ship-based time-series network by collaborating closely with the participating time-series programs, developing, and recommending SOPs, and supporting the network to become more fit-for-purpose.



## 9. Outlook

We envision the SPOTS pilot to be the basis for a sustained living data product of time-series data that supports the timely delivery of scientific information on ocean biogeochemistry trends and variability across the main bio-eco domains of the world ocean. A product that complements SOCAT, GLODAP, and MEMENTO, together forming the primary source of EOV data for global marine BGC research and assessment. Three related near-term goals would be to i) regularly update the data of the already included sites to extend the data coverage in time; ii)

extend the product by attracting further ship-based time-series programs measuring BGC EOVs, linking to the plus 340 sites identified by IGMETS and particularly closing the gap in the Indian and Southern Ocean to extend the data coverage in space; and iii) promote further development and adoption of BPs and the proposed SOPs by the ship-based ocean time-series community. In the long term, the product could extend the pilot's scope beyond BGC EOV data and include biological EOVs, as well as measurements from moorings. Work towards a "bio use-

case", initiated by METS RCN, has already started, and leveraged from the knowledge and methods developed by this pilot effort. Including time-series data from moorings is far beyond current capabilities though. More generally, we hope that this effort contributes to increasing the recognition of the utility and value of ship-based BGC time-series data. A (ship-based) time-series BGC observing network that collaborates with the observing programs Surface Ocean $CO_2$ Reference Observing Network (SOCONET) and GO-SHIP and that complies with

the requirements of an ocean observing network, as articulated by the GOOS Observations Coordination Group, needs to be established accordingly. This network should govern the product using an integrated approach with the existing BGC data synthesis products SOCAT and GLODAP.



**Author contributions**

NL, TT, and BF led the team that produced the SPOTS pilot. ABC, AW, BF, DRR, EH, FMK, FP, IS, KC, LC, MA, MH, MW, NL, SF, and SRO compiled and/or provided original datasets and metadata. FMK, DRR, KC, NL, SL, and TT developed the BGC EOV metadata template. NL, TT, and BF developed the recommended QC guidelines for BGC EOV time-series programs. NL conducted additional QC analyses on a few original datasets, executed the assessments (method evaluation, minimum variability, GLODAP offset), merged the SPOTS pilot

and applied the TOATS notebook to the data. PLB and NL generated machine-readable metadata (ODIS) for all time-series programs and the SPOTS pilot itself. PLB manages the inclusion of the data product and individual time-series programs into the ODIS catalog and ODIS user interface. KOB uploaded the data product to ERDDAP and implemented all related functionalities. HB coordinates METS RCN and related activities (website, workshops). All authors contributed to the manuscript.


**Competing interests**

The authors declare that they have no conflict of interest.

**Acknowledgments**

The SPOTS pilot would not have been possible without the effort of the many scientists who secured funding, dedicated time to collect data, and shared the data that are included. Principal investigators of the different stations are listed on the METS RCN website. The development of this pilot product was supported by the International Ocean Carbon Coordination Project (IOCCP) by valuable input from its scientific steering group members as well as from the OCB Time-Series Panel. Further, we want to acknowledge Adrienne Sutton for leading the

development of TOATS and contributing to Sect. 6.2.

**Financial Support**

NL was funded by EU Horizon 2020 through the EuroSea Innovation Action (grant agreement 862626). BF was funded by the WASCAL MRP-CCMS project from the German Federal Ministry of Education and Research

(BMBF; grant agreement no. 01LG1805A). The work of SRO was supported by the European Union's Horizon 2020 research and innovation program under grant agreement no. 820989 (COMFORT). FMK and DRR from the CARIACO time-series project were supported financially by the National Science Foundation (OCE-1259043). The work of IS was supported by Norwegian Environment Agency under grant agreement nos. 14078029, 15078033, 16078007, 17018007, and 21087110. The work of K2 and KNOT was partly supported by a Grant-in-

Aid for Scientific Research (20H04349) from the Ministry of Education, Culture, Sports, Science, and Technology (MEXT) KAKENHI. The DYFAMED ship-based time series is supported by the MOOSE program (Mediterranean Ocean Observing System for the Environment) coordinated by CNRS-INSU and the Research Infrastructure ILICO (CNRS-IFREMER). GIFT has been supported by the European projects CARBOOCEAN, CARBOCHANGE, SESAME, PERSEUS and COMFORT, the Spanish Ministry of Science through the grants

CTM2005/01091-MAR and CTM2008-05680-C02-01 and the Junta de Andalucia through the TECADE project (PY20_00293). The work of HOT is supported by the NSF (OCE-0926766). The METS RCN is supported by NSF 2028291.



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
