# Peer review of "Synthesis Product for Ocean Time-Series (SPOTS) – A ship-based biogeochemical pilot"

_Earth System Science Data, 2023_

## Author Response (AR1)

**Reviewer #1**

**Reviewer Comment #1**

There are still some other time series measurements of carbonate chemistry stations such as Hawaii Ocean Time Series (HOT), Bermuda Atlantic Time Series Study (BATS), European Station for Time Series in the Ocean Canary Island, (ESTOC) and South East Asia Time Series study (SEATS). Since this paper provides carbonate chemistry data, why not incorporate the data from these stations into the study? The authors may need to consider adding these time-series datasets this paper.

**Author's response**

The HOT data is included in the pilot, however the other station data could not be included due to different reasons. Each of these time-series was contacted and asked for participation in the pilot though.

ESTOC: The ship-based (distinct) data was not ready in time preventing a possible inclusion of these very valuable data in this pilot product. Very recently ESTOC was able to publish parts of their bottle data (carbonate chemistry, with temperature and salinity; https://www.frontiersin.org/articles/10.3389/fmars.2023.1236214/full) covering a time from 1994 until 2023. However, there is still work ongoing regarding their nutrient measurements. Contact with the PIs of ESTOC has been established since the beginning of this project and is still ongoing so that for the planned update of SPOTS, the inclusion of ESTOC`s full bottle datasets are a priority.

BATS: At the early stages of this effort (concept phase), the scientific team behind BATS have informed us that they did not want to be part of this pilot. Despite several discussions we could not convince them to participate. However, we hope that through the publication of this pilot we can convince them of the importance of SPOTS and include BATS in the future.

SEATS: We contacted SEATS together with the other included time-series stations. Unfortunately, we did not receive a reply.

Despite these (and other) time-series data missing in this pilot of SPOTS, we believe that the pilot represents the fundament for a larger, living, and sustained SPOTS in the near future. We further believe that the above listed stations will be included with the upcoming updates.

**Changes in manuscript**

None.

**Reviewer Comment #2**

Table 3. The desired part is not clear to me. Authors may need to consider explaining more details about the method used for each parameter and providing appropriate references for those methods.

**Author's response**

We added a supplementary table, S9, which links the used references for defining required and desired method recommendations to each variable. Note that in some instances we also relied upon internal discussions and comparisons that are not published. Such instances are indicated accordingly, too. We additionally added an example in Section 3.3.2 (using total alkalinity) to explain the meaning of Table 3, the mentioned methods, and the corresponding flag assignment, in more detail.

Further, the parameters "Particulate Matter" refer to TPN, TPP, and TPC – the manuscript (and product) has been updated accordingly.

**Changes in manuscript**

Added example in Section 3.3.2 (Lines 362 – 382).

**Reviewer comment #3**

Line 135 Table 1: NO3- indicates dissolved inorganic nitrate and NO2 indicates dissolved nitrite. Please be consistent in the text.

**Author's reply**

We updated the manuscript accordingly so that it is consistent.

**Changes in manuscript**

Updated in Line 138.

**Reviewer comment #4**

Line 225 to 226: The Table 1 shows that K2 and KNOT started in 1999 and 1997, respectively, which is not consistent with the information in your main text Line 225 to 226.

**Author's reply**

Actually, both dates haven't been correct. All wrong instances were corrected.

**Changes in manuscript**

See changes in Table 1 and Line 229.

**Reviewer comment #5**

Line 300: Typo in Lauvest and Tanhu, 2015))

**Author's reply**

Typo not found.

**Changes in manuscript**

None.

**Reviewer comment #6**

Line 355 Table 3. In the variable column, is parameter referring to POC, PON, POP?

Also, carbon and nitrate. Isn't it carbon and nitrogen? Should clarify the parameters related to particulate matter.

**Author's reply**

All particulate matter terms now refer to total particulate matter. And, yes, it is nitrogen instead of nitrate.

**Changes in manuscript**

Table 1, 3, 6 updated, as well as all instances within the text (multiple) and headers in Figure 3.

**Reviewer comment #7**

Line 365-385: What about salinity, particulate carbon and particulate nitrogen?

**Author's reply**

Precision estimates of salinity, TPC, TPN and TPP that are based on replicate measurements are included in the product. Note that for particulate matter neither CVOO nor CARIACO provide such measurements. The latter provides precision values of the instruments which would not provide a fair comparison to precision estimates of replicate samples though. Similarly, all existing accuracy measurements of particulate carbon and nitrogen are either based on instruments (CARIACO) or ratios (TPC/TPN; ALOHA) and were not included in the product as only accuracies based on measurements against reference materials were included. Salinity accuracy estimates were not provided.

**Changes in manuscript**

None.

**Reviewer comment #8**

Line 372-373: Please provide more specific information about which carbon variable you are referring to DIC, DOC or POC?

**Author's reply**

The inorganic carbon variables are meant.

**Changes in manuscript**

Line 401 updated.

**Reviewer comment #9**

Line 580-581: Why not use the estimated partial pressure of CO2 data from DIC, TA and pH measurements?

**Author's reply**

We have decided to only include measured variables in the product. There are several reasons for this decision, one of them being a clear link between original data and the synthesis product. However, in particular regarding calculating missing inorganic carbon parameters, recent studies indicate that measured and calculated variables can differ significantly, and the estimations also depend strongly on the variables used (e.g., using DIC and TA to calculate pCO2 vs using DIC and pH to calculate pCO2). Accordingly including such calculated variables might harm/bias trend analyses, especially if combining measured data from different sources, in terms of analytical and quality control procedures.

Two review works are being prepared for publication from the OCSIF OCB working group, https://www.us-ocb.org/ocean-carbonate-system-intercomparison-forum/, dealing with the current state and recommendations for CO2 system measurements, the caveats and future work on internal consistency studies and data analysis and reporting. Particular recommendations and warnings about combining measured and calculated CO2 data are given. The main message is that a careful evaluation of the uncertainties in both measured and calculated data should be given to finally commit any trend analysis.

We have chosen to suggest methods on how to increase the data coverage through computations in Section 6.2 (e.g., Lines 861 – 867).

**Changes in manuscript**

None.

**Reviewer comment #10**

Line 808: Typo in de Boyer Montégut et al. (2004))

**Author's reply**

Typo not found.

**Changes in manuscript**

None.

**Reviewer comment #11**

What does NA mean on table 4 and 5?

**Author's reply**

NA stands for Not Available.

**Changes in manuscript**

Explanation for this abbreviation included in table captions.

**Reviewer # 2**

**Reviewer comment #1**

First, the authors didn't include some other easily accessible time-series stations in the current compilation, such as BATS, OSP, and ESTOC. The data product would be more complete if the authors could add those as well. See Figure 2 in Benway et al. (2019) (https://doi.org/10.3389/fmars.2019.00393) for the locations of these stations.

**Author's reply**

ESTOC: The ship-based (distinct) data was not ready in time preventing a possible inclusion of these very valuable data in this pilot product. Very recently ESTOC was able to publish parts of their bottle data (carbonate chemistry, with temperature and salinity; https://www.frontiersin.org/articles/10.3389/fmars.2023.1236214/full) covering a time from 1994 until 2023. However, there is still work ongoing regarding their nutrient measurements. Contact with the PIs of ESTOC has been established since the beginning of this project and is still ongoing so that for the planned update of SPOTS, the inclusion of ESTOC`s full bottle datasets are a priority.

BATS: At the early stages of this effort (concept phase), the scientific team behind BATS have informed us that they did not want to be part of this pilot. Despite several discussions we could not convince them to participate. However, we hope that through the publication of this pilot we can convince them of the importance of SPOTS and include BATS in the future.

OSP: OSP is clearly a very important fixed time-series station with a long record including biogeochemical data. However, we chose not to include OSP in the pilot of SPOTS as all DFO Line-P data was recently embedded in GLODAP. Having said that OSP data will be a priority for an update of SPOTS which will become even more important once mooring data are included in SPOTS as well.

Despite these (and other) time-series data missing in this pilot of SPOTS, we believe that the pilot represents the fundament for a larger, living, and sustained SPOTS in the near future. We further believe that the above listed stations will be included with the upcoming updates, as hopefully accompanied by many others.

**Changes in manuscript**

None.

**Reviewer comment #2**

Secondly, I think one aspect of future perspectives that could be emphasized more. Other BGC parameters are also routinely measured at some of the time series stations. For example, parameters such as pigments, dissolved organic nitrogen (DON), and sediment trap particles are measured at Station ALOHA. Likewise, DON, dissolved organic phosphorus, and sediment trap particles are measured at CARIACO.

**Author's reply**

We added an extra sentence in the outlook section, now also extending the SPOTS vision to BGC non-EOVs. However, for the pilot we are convinced that focusing on BGC EOVs is appropriate following the focus set by the Framework of Ocean Observations.

**Changes in manuscript**

Updated lines 1046-1048.

**Reviewer comment #3**

Lines 135–136: Which time-series station did you use CTD rather than bottle oxygen data?

**Author's reply**

All included data are bottle data.

**Changes in manuscript**

Caption of Table 1.

**Reviewer comment #4**

Line 272- Figure 2: Why is applying QC tests optional? Based on 3.3.1, QCs from the time-series stations are applied if they are available. To me, it seems like the authors are applying this step to all stations before the BP checks. Therefore, it's not an optional step, and the authors should consider removing "(optional)" here.

**Author's reply**

This step relates to "external" QC, i.e. (additional) QC applied by the SPOTS team (in collaboration with the time-series program). Only Munida and CVOO have chosen to undergo this external check, all other flags have been assigned by the time-series programs themselves, i.e., "internally".

**Changes in manuscript**

Figure 2 is updated accordingly to highlight that external QC is meant here.

**Reviewer comment #5**

Lines 276–277: Are data from Munida and RADCOR directly obtained from the responsible PIs? If so, the authors should either state it here or in Table 1 or S1. In some cases, the data downloaded from data centers do not cover all years mentioned in Table 1. Did the authors get those missing data directly from PIs? Please clarify.

**Author's reply**

The data sources are provided in S1. The last column (G) shows "PI" if the complete and merged data is obtained directly from the PI (e.g., Munida). The entry "Both" indicates that some data are already archived (i.e., a DOI is present) and that missing/additional data was obtained directly from the PI. Note that for some time-series programs the data was obtained from multiple published data sources, e.g., IrmingerSea, KNOT, RADCOR. For those instances all corresponding DOIs are provided in column F in S1. We added an extra sentence to make this clearer.

**Changes in manuscript**

Updated lines 283-284.

**Reviewer comment #6**

Lines 304–305: If the authors are converting the carbonate system parameters (e.g., pH) that are sensitive to temperature, it makes sense to use reported laboratory temperature for the conversion. However, for nutrients and DOC, it makes more sense to use in-situ T and S measurements for the conversion.

**Author's reply**

We followed the recommendations by Jiang et al, (2022), who explicitly state that for nutrients that the lab-temperature should be used. The reason behind is the following: given the methodology for the nutrient analysis, usually segmented flow analysis, all the solutions (samples and reagents) are assumed to be stable and at the lab temperature that should be constant.

**Changes in manuscript**

None.

**Reviewer comment #7**

Lines 311–312: The conversion of POP from ug/kg to umol/kg should also be mentioned here.

**Author's reply**

TPP was given in nM (CARICAO, see S4) and thus there was no need to use/mention the inverse standard atomic masses of phosphorus.

**Changes in manuscript**

None.

**Reviewer comment #8**

Lines 313–314: If the authors look at sediment trap data at Station ALOHA which have measurements of both total PC and PIC flux, one can calculate the fraction of PIC in sinking particles. The mean fraction of all data is 10.4±4.5% using all the data. If we make the assumption that suspended particles collected by bottles have the same PIC% as sinking particles, we can easily see that the authors would overestimate the POC concentrations at Station ALOHA by ~11%. This is not a fraction that can be ignored, even at the subtropical Station ALOHA. The authors should consider labeling these data (e.g., PC as POC) with different quality flags to make users understand this caveat or incorporate this in the uncertainty analysis.

**Author's reply**

All particulate matter terms now refer to total particulate matter.

**Changes in manuscript**

Table 1, 3, 6 updated, as well as all instances within the text (multiple) and headers in Figure 3.

**Reviewer comment #9**

Lines 450–455: Do the authors use samples from all depths or just below 1500 db? It's a bit unclear what the authors meant by relaxing the minimum depth requirement.

**Author's reply**

The column 'Layer' in Table 5 shows the layers used for the crossover analysis, in most cases also explicitly mentioning the related minimum depth.

**Changes in manuscript**

Updated lines 484-485.

**Reviewer comment #10**

Line 624: Does this high fraction also represent the natural variabilities of DOC at Station ALOHA? If it's not circulation-driven (since low variability in T and S), is it possible to be caused by biological activities? The same question can be applied to all parameters that have a higher variability than the accuracy of the method.

**Author's reply**

Generally, if circulation effects can be excluded through low salinity (and temperature) variabilities, it is reasonable to link relatively high coefficients of variation (in comparison to accuracy) estimates to biological activity. However, to minimize the effects of biological activity we have chosen to estimate the coefficients of variabilities on the most invariable oxygen layers (Lines 415 – 420). Given that for, e.g., ALOHA the coefficient of variation for oxygen is 0.7%, we find it unlikely that the high variabilities of DOC are mainly dominated by biological effects, but rather represent measurement uncertainty not fully captured by the usage of RMs or blank control. The same argument holds for other relatively high coefficients of variations. Also note, that ALOHA accuracy estimates are only provided for four years (2008 – 2011). Nevertheless, even if oxygen variations indicate an established equilibrium, we cannot fully exclude that biological activity can have effects on the calculated variabilities of the other BGC measurements. In the revised version, we have highlighted that users must cautiously interpret these results.

**Changes in manuscript**

Updated lines 664.

**Reviewer comment #11**

Lines 634–635: It would be helpful if the authors could list the names of all these stations so that we can determine where to see high natural variabilities.

**Author's reply**

Done.

**Changes in manuscript**

Updated line 669.

**Reviewer comment #12**

Line 660: What about 33KB20020923 or 325019971101 or 49HG20010813? They seem to fall within the criteria, at least for the 2˚ spatial box, unless the minimum of 1500 m criterion is applied.

**Author's reply**

Given the large pool of deep data at ALOHA, we have chosen to set the minimum depth to 2000 dbar (Table 5, column "layer"), but still at least two samples from the same profile are needed below that depths to perform the crossover analysis. For the mentioned and not considered cruises the minimum depths would actually have to be below 1000 dbar.

**Changes in manuscript**

None.

**Reviewer comment #13**

Line 955: I may have missed it. Is this point about O2 and pH discussed at all in the manuscript? If not, please add more detail.

**Author's reply**

It is discussed in Sections 4.1.2 and 4.1.6, respectively.

**Changes in manuscript**

None.

**Reviewer comment #14**

Line 66: Please double-check this doi number. I cannot find this dataset using this doi.

**Author's reply**

The DOI should work now.

**Changes in manuscript**

None.

**Reviewer comment(s) #15**

Table 1- Variables: Why not include temperature as one of the measured properties? Some of these stations have sediment trap measurements, such as Station ALOHA, CARIACO, and BATS.

Table 1- KNOT: Data downloaded from this link is from 1992 to 2008, not 1997 to 2020. Plus, there are no direct measurements of pH reported.

Table 1-K2: Data are only from 1999 to 2008. Neither pH nor DOC values are reported.

Table 1-ALOHA: Data are from 1988 to 2016 rather than 1988 to 2019. Plus, one can download data from Station ALOHA from 1988 to the end of 2022 for the bottle data. Why do the authors only report time ranges between 1988 and 2019?

Table 1-Munida: Are data from the Munida and RADCOR not available?

Table 1-GIFT: There aren't any PO4 and DOC data available associated with the DOI.

Table 1-CVOO: The dataset is currently in review and the DOI link doesn't work now.

Table 1-Irminger Sea: The current time range is from 1983 to 2019, but there are also data available from 2020 to 2022.

Table 1-Iceland Sea: Likewise, there are also measurements from 2020 to 2022. Additionally, the earliest measurements were taken in 1985 rather than 1983.

**Author's reply**

All typos/errors are fixed. Further note that

- 'NA' only denotes that no DOI is available for the corresponding datasets.
- The ALOHA data we used are the most recent one with a DOI (from BCO-DMO, https://www.bco-dmo.org/dataset/3773#data-files, see Suppl. Table 1).

- A DOI has been added to the RADIALES A Coruña (RADCOR) times series
- A DOI has been added to the GIFT time series
- The Irminger and Iceland Sea time-series additional data (2019-2022) were archived / provided too late for their inclusion into the pilot (see 'Lineage' at metadata landing page).

**Changes in manuscript**

Updated Table 1.

**Reviewer comment(s) #16**

Line 218: Replace "IS-TS" with "IC-TS".

Line 290: Replace "fill" with "filling".

Line 298: The citation should be Jiang et al. (2022) rather than Liqing et al. (2022). Change it throughout the manuscript.

Line 302: In Table S4, particulate matter has the unit of ug/kg in the tab "Product" and umol/kg in the tab "Time-series Stations". Be consistent and change all units to umol/kg.

Line 356- Table 3: Replace "All" with "All except silicate".

Line 480: Replace "E.g." with "For example".

**Author's reply**

Done.

**Changes in manuscript**

Updated throughout the manuscript.

**Reviewer comment #17**

Line 565: It would be helpful to have figures like Figure 4 or 5 for all parameters in the supplemental, not just for nitrate and TA.

**Author's reply**

A set of figures (one for each time-series) has now been attached in the supplemental.

**Changes in manuscript**

None.

**Reviewer comment(s) #18**

Lines 638 & 653: Use "Hash" rather than "Rhombus".

Line 653 & 859: Use "Asterisk" rather than "Asterix".

Line 654: Is it just CTD salinity or CTD oxygen? If it's just CTD salinity, please be more specific.

**Author's reply**

Done.

**Changes in manuscript**

Updated throughout the manuscript.

**Reviewer comment#19**

Tables 4&5: Add a column of temperature.

**Author's reply**

GLODAP has not (yet) implemented temperature offset nor does it provide consistency estimates for temperature. We have accordingly only added temperature to Table 4 and not to Table 5.

**Changes in manuscript**

Updated Table 4.

**Reviewer comment #20**

Line 654: The authors should consider reporting the mean offset consistently either as an absolute number or a relative fraction, rather than a combination, in Table 5.

**Author's reply**

The offsets are given following the example set by GLODAP, which uses a combination of both types depending on the range of expected values.

**Changes in manuscript**

None.

**Reviewer comment #21**

Line 654-IcelandSea: Why is there only one number in the parathesis? If it's either 3:0 or 0:3, how can the authors get 1% for the comparison?

**Author's reply**

Typo corrected.

**Changes in manuscript**

Updated Table 5.

**Reviewer comment(s) #22**

Line 705: Suggest separating the dashed line and the text more in Figure 6.

Line 855: Consider using black-outlined empty circles for the monthly means so that observations can be better seen.

Line 860: If the fit is not significant, I don't think the authors need to put a red line fit. It will make it easier for the readers to see which relationships are significant.

**Author's reply**

Both figures are generated by existing scripts and tools, accordingly we have not updated their layout.

**Changes in manuscript**

None.

**Reviewer comment #23**

Line 891: The data product is not available online yet. The authors need to post the data product before this manuscript is accepted.

**Author's reply**

Done.

**Changes in manuscript**

None.

**Reviewer comment(s) #24**

Line 906: This link doesn't work. Is it supposed to be
"https://data.pmel.noaa.gov/generic/erddap/tabledap/spots_bgc_ts.html"?

Line 915: This link doesn't work. Is it supposed to be
"https://data.pmel.noaa.gov/generic/erddap/tabledap/spots_bgc_ts.graph"?

**Author's reply**

Yes.

**Changes in manuscript**

Lines 948 and 958-959 updated.

---

## Author Response (AR2)

We want to thank the reviewers for their thorough examination of the manuscript and dataset. Below you can find our detailed reply to the extra comments by Reviewer #2 which helped us in erasing the last shortcomings.

Please note, that these comments also affected the dataset itself (extra columns for particulate organic matter) and the Supplementary files. The updated dataset is presently under revision at BCO-DMO, the corresponding DOI has however already been provided and updated in the manuscript, too.

**Reviewer Comment #1:**

Lines 140–141 (Table 1): Not all time-series stations measure the same type of particles. Some of them measure the concentrations of total particles, while others only measure organic particles. Therefore, it's not reasonable to use total particulate carbon (TPC), nitrogen (TPN), or phosphorus (TPP) to describe particulate measurements across different stations. For example, CVOO measures POC, PON, and POP rather than TPC, TPN, and TPP. Likewise, the authors need to label particulate measurements with the correct names in Figure S10 for different time-series stations.

**Reply to Comment #1:**

After consultation with the PIs of CVOO, we agree that particulate matter measured at CVOO should not be represented as total particulate matter but as exclusively organic particulate matter. We have updated figures, tables and the manuscript accordingly. We have also included the reason for this distinction to the measured particulate at ALOHA and CARIACO in line 618 – 622, which "also point to the readers the difference between total particulate matter and particulate organic matter" (see Reviewer comment #3).

**Reviewer Comment #2:**

Line 280 (Figure 2): The authors cannot do any offset analysis with particulate data since GLODAP has no particle data. In this case, the authors should consider updating this schematic with two scenarios, one for GLODAP core variables and the other for other EOVs.

**Reply to Comment #2:**

Since all offsets are linked to GLODAP, we feel only confident in showing offsets to QC'ed data, i.e., to the core parameters of GLODAP.

**Reviewer Comment #3:**

Line 523 (Figure 3) and Line 786 (Table 6): It may work better to use "TPC (POC)", "TPN (PON)", and "TPP (POP)" in the subplot titles and column names. The authors should double-check their descriptions/discussion about particles throughout the manuscript and also point to the readers the difference between total particulate matter and particulate organic matter. This is important so that the users can use the data correctly.

**Reply to Comment #3:**

We have updated the Figure and Table accordingly. However, we haven't changed the subtitles, but instead have added extra sentences in the figure/table captions. All instances in the manuscript have been checked, as well, and are consistent (total particulate matter for ALOHA and CARIACO; particulate organic matter for CVOO).

**Reviewer Comment #4:**

Lines 1048–1049: How do the authors define "BGC non-EOVs" here? Some of the future measurements to include can also be BGC Essential Ocean Variables. In my comment from the last round, I suggest including measurements such as DON or sediment trap flux in future work. Here, the authors generally categorized those as the BGC non-EOVs. I don't think it's accurate. For example, sediment trap particle fluxes, such as POC, CaCO3, and biogenic silica flux, are included in the particulate matter EOV summarized by GOOS. Although DON is currently not listed as one of the EOVs, it belongs to dissolved organic matter in general and can also be used as dissolved nutrients in the nutrient-depleted subtropical gyre (e.g., Letscher et al., 2016). This reviewer is not very satisfied with the response to this comment from the last round. Suggest rephrasing this sentence with more accurate definitions.

**Reply to Comment #4:**

We have specified the present focus defined by **bottle** BGC EOVs (line 132) and added an extra sentence about the possible extension towards sediment traps derived data (line 133 – 134).